# Language models are weak learners

**Hariharan Manikandan**[1]    **Yiding Jiang**[1]    **J Zico Kolter**[1,2]
[1]Carnegie Mellon University    [2]Bosch Center for AI
{hmanikan, yidingji, zkolter}@cs.cmu.edu

## Abstract

A central notion in practical and theoretical machine learning is that of a *weak learner*, classifiers that achieve better-than-random performance (on any given distribution over data), even by a small margin. Such weak learners form the practical basis for canonical machine learning methods such as boosting. In this work, we illustrate that prompt-based large language models can operate effectively as said weak learners. Specifically, we illustrate the use of a large language model (LLM) as a weak learner in a boosting algorithm applied to tabular data. We show that by providing (properly sampled according to the distribution of interest) text descriptions of tabular data samples, LLMs can produce a summary of the samples that serves as a template for classification and achieves the aim of acting as a weak learner on this task. We incorporate these models into a boosting approach, which in some settings can leverage the knowledge within the LLM to outperform traditional tree-based boosting. The model outperforms both few-shot learning and occasionally even more involved fine-tuning procedures, particularly for tasks involving small numbers of data points. The results illustrate the potential for prompt-based LLMs to function not just as few-shot learners themselves, but as components of larger machine learning pipelines.

## 1 Introduction

Weak learners refer to classifiers that are able to attain better performance than random chance, by some given margin, on any specified distribution over training data. One of the early breakthroughs in machine learning established that this weak learning was sufficient for arbitrarily strong classification, via an ensembling procedure [Schapire, 1990]. This approach in turn led to the development of boosting algorithms [Freund and Schapire, 1997], a class of approaches that continue to perform extremely well, particularly on tabular datasets that lack the input space regularity of vision or language tasks.

In a seemingly separate thread of research, large language models (LLMs) based on transformers [Vaswani et al., 2017] in recent years have come to dominate many natural language domains. These models are often finetuned on the data of new downstream tasks [Devlin et al., 2018, Liu et al., 2019], but in recent years have also been shown to exhibit strong performance as zero-shot or few-shot learning solely via prompting the model [Brown et al., 2020] with a piece of context string.

In this paper, we align these two threads of research and ask a simple question: *can LLMs also serve as weak learners in a boosting framework, specifically on tabular data (where boosting methods are most commonly applied)*? We answer this question largely in the affirmative. Specifically, we show that by appropriately converting tabular data to text form, and asking LLMs to summarize a carefully chosen set of examples from the data, we produce a summary of the examples that can serve as a template (i.e., a prompt) for a tabular data classifier, and one which typically achieves this weak learning aim. This enables us to correspondingly integrate this collection of LLM-generated weak learners into a boosting framework.

37th Conference on Neural Information Processing Systems (NeurIPS 2023).

We show that the resulting approach performs well in many settings, easily outperforming zero-shot and few-shot classification, as well as "single-shot" summaries generated by the LLM. This is all done without any retraining or finetuning of the LLM itself, but rather only via prompting. Furthermore, on certain domains (particularly those with very few examples, where leveraging the prior knowledge built into LLMs would be of particular importance), we show that the approach can even outperform traditional tree-based boosting and LLM-based finetuning methods and its performance would likely improve as LLMs capabilities improve. Overall, we believe this work highlights the potential of incorporating LLMs as sub-routines of a larger machine learning system.

## 2   Related Works

**Deep Learning for Tabular Data.**   Tabular data refers to a generic data format that represents data as a collection of discrete or continuous attributes [Borisov et al., 2021]. Due to their flexibility, tabular data are ubiquitous in many ML settings. However, such flexibility comes with a cost – they lack the inherent structure found in images or text, which makes applying deep learning to them challenging. Furthermore, they are often domain-specific and may have a relatively small number of data points. As a result, traditional deep learning methods, which thrive on large datasets and high-dimensional data, have seen limited success when applied to tabular data [Gorishniy et al., 2021, Shwartz-Ziv and Armon, 2022].

Recently, however, there has been increasing interest in applying deep learning to tasks related to tables such as data integration, imputation [Narayan et al., 2022], semantic parsing, and even running SQL queries [Herzig et al., 2020, Yin et al., 2020]. Deep learning models have also been successful at learning tabular data classification by optimizing loss functions [Hollmann et al., 2022, Schäfl et al., 2022, Dinh et al., 2022]. Unlike these approaches, we study how we can use LLM for classifying tabular data *without* finetuning or building a new language model. Since many tabular data can be grounded in natural language, texts are in fact a *natural representation* for tabular data. Motivated by the observation that LLMs can convert tables to text through prompting alone [Saha et al., 2022], we utilize LLMs to do this conversion. After the conversion, our classification algorithm also interacts with existing LLMs strictly through prompts. This creates an abstraction between the underlying language model and the learning procedure which may be desirable for various applications since access to the gradients or parameter updates are not required.

**Prompting**   Prompting [Liu et al., 2023] refers to providing initial text or instructions to guide the response of a language model. The advancements in Language Model-based Learning (LLM) have unveiled new capabilities, such as chain of thought reasoning [Wei et al., 2022], zero-shot reasoning [Kojima et al., 2022], compositional problem solving [Zhou et al., 2022a], and self-improvement [Huang et al., 2022, Ho et al., 2022, Haluptzok et al., 2022]. As a result, prompting has gained widespread application across various Natural Language Processing (NLP) tasks, including arithmetic, common reasoning [Wang et al., 2022b], among others [Brown et al., 2020]. While prompts offer flexibility, it is crucial to note that LLMs interpret them differently from humans. Therefore, the process of *prompt tuning*, which involves carefully engineering prompts, becomes essential for obtaining accurate and relevant outputs [Reynolds and McDonell, 2021]. At its core, prompt tuning is an optimization process that aims to find the best prompt for a certain downstream task. Though a long line of works propose gradient-guided search to optimize "continuous prompt" instead of the language tokens [Liu et al., 2021, Qin and Eisner, 2021, Lester et al., 2021, Shin et al., 2020, Rakotonirina et al., 2023, Wang et al., 2022c, Diao et al., 2023], gradient-based updates can be limiting, as LLMs become bigger and the access to these models become increasingly API-based. Our approach aligns more with discrete search methods based on the fact that LLMs can automatically generate prompts for themselves [Zhou et al., 2022b, Zhang et al., 2022, Yu et al., 2022]. Specifically, we prompt the LLM to summarize the tabular dataset. The summary in turn acts as a prompt that the LLM uses to make predictions as it encodes knowledge of the dataset. A sequence of such prompts summarizing different subsets of the data can be seen as weak learners for a boosting procedure.

**Boosting**   Boosting [Schapire, 1990, Freund and Schapire, 1997] is a widely used technique to improve the accuracy of a model by combining weak learners (models that perform slightly better than random guessing) to make a strong learner (model with high accuracy). Common boosting algorithms include AdaBoost [Freund and Schapire, 1997], Gradient Boosting [Friedman, 2001], and Stochastic Gradient Boosting [Friedman, 2002]; the XGBoost library [Chen and Guestrin, 2016] in

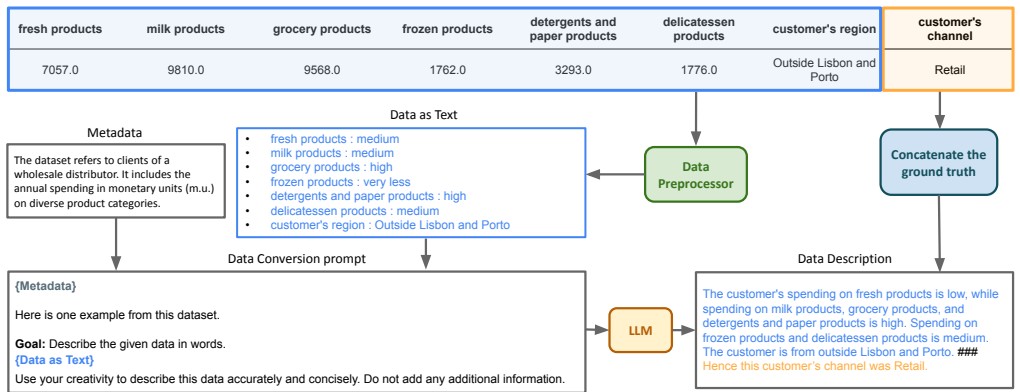

Figure 1: The conversion for a data point on the Wholesale customers dataset (OpenML ID 1511).

particular is a commonly used implementation of gradient boosting. In concurrent work most relevant to ours, Hou et al. [2022] integrate LLM into AdaBoost for natural language inference (NLI) tasks, by training an MLP projection of the final hidden state of a special token. Our proposed method diverges from theirs in two key aspects. Firstly, our method avoids the overhead of learning additional parameters, is gradient-free, and does not require access to the model's internal states. Secondly, instead of storing knowledge in parameters, our approach concentrates on condensing knowledge into an intermediary representation referred to as "summary." This alternative strategy enhances interpretability and strictly learns through prompts, rendering it particularly suitable for small tabular data, where the prior knowledge in LLM can significantly benefit the learning process.

## 3 Summary Boosting with Language Models

We now describe the main methodology of our paper, which uses LLMs to generate weak learners, and in turn, uses these weak learners within a boosting framework. We refer to the full method as *Summary Boosting*, as the core learning process is one that uses a language model to create a summary of (specifically chosen) samples from the dataset; these summaries themselves function as prompts by which we can make predictions on new examples. Finally, we use boosting to construct an ensemble of these summaries that gives the overall predictions on new data points.

### 3.1 Data conversion

To utilize large language models (LLMs) with tabular data, it is necessary to first convert the records into natural language descriptions. We will refer to these as *data descriptions*. Template matching, commonly used in previous approaches [Dinh et al., 2022], inserts attribute values into predefined templates. However, this approach often produces unnatural descriptions that differ from how humans might describe the data. Depending on the dataset, designing the template by hand can also be challenging. To overcome this, we propose using LLMs as a more suitable solution.

As illustrated in Figure 1, we can get these data descriptions with little effort by *zero-shot prompting* the LLM with information about the dataset (which is generally available as metadata for tabular datasets) and a textual representation of the tabular record (e.g., parsed JSON). Specifically, to ensure examples can serve as both training data and query inputs, we extract the descriptions of the features and concatenate them with the target label using a separator token (refer to Appendix A.1). Interestingly, we find that descriptions generated by LLM this way often perform better than those from a template. This ablation study can be found in Section 5.

One key challenge in this process is how to encode numerical attributes effectively; naively including numerical values in the descriptions can lead to poor performance in subsequent learning tasks. To address this, we adopt a straightforward approach: we bin all numerical features into percentiles and encode them descriptively as "low," "medium," and "high,". In Section 5, we compare the performance of several such approaches and discuss more examples in Appendix A.6. Overall, the data descriptions can be generated automatically with *minimal manual engineering*.

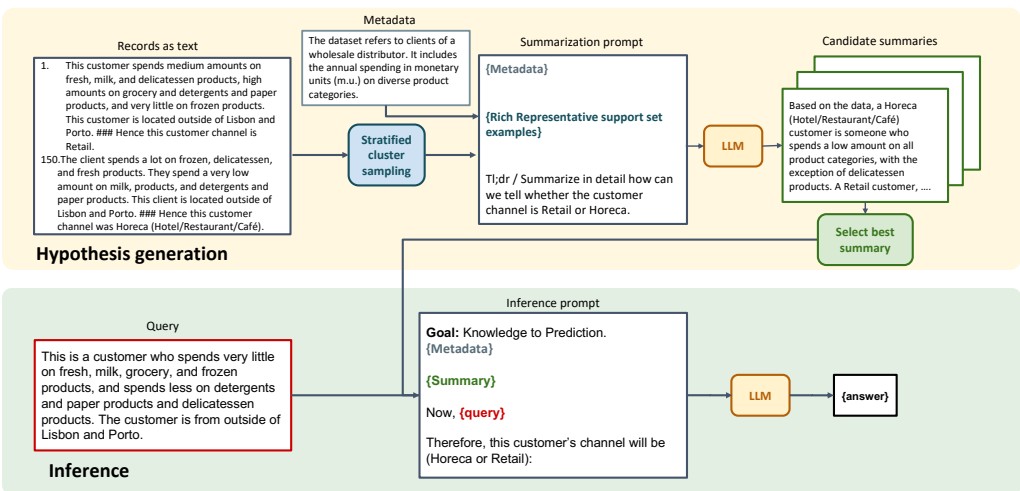

Figure 2: The process of generating summaries and using them to make predictions on new data. The top half describes how the weak learning hypothesis (summary) is generated. The bottom half illustrates how the summary is used to perform inference. "Best summary" is the one achieving the smallest validation error rate, or higher training error in case of a tie.

## 3.2 Weak learning via summarization

A typical method for performing few-shot learning with large language models (LLMs) involves providing a small number of demonstrations of the intended task as a prompt and then asking the model to generate an answer. One could, for instance in the few-shot setting, simply present the natural language descriptions above and generate predictions on new examples. However, for tabular data, there may be a larger number of data points that do not fit within the LLM context. Furthermore, we observed that increasing the number of examples in the context naively does not always improve performance (Figure 4, right bottom), and there was no obvious way to manage weighted distributions over examples as is required in boosting methods. These observations necessitate alternative approaches to weak learning via LLMs.

We propose instead that *producing summaries of a collection of examples* can serve as a powerful proxy for learning models based upon some number of examples, as summarization naturally encourages the extraction of representative information in the data. Concretely, given a set of data descriptions, we first perform summarization on the data by calling the LLM (e.g., by concatenating a list of examples in natural language form and appending the prompt "tldr"). This resulting summary can be seen as a hypothesis as it provides an explanation for the data. By using the summary as a prompt, the LLM in turn uses the hypothesis to perform inference instead of the raw data description (shown in Figure 2). Since the sampled summary can sometimes be noisy, we generate a fixed number of summaries and pick the one with the smallest validation error rate. In case of a tie, we choose the one with a higher training error, i.e., a lower generalization gap (see Appendix A.2). Several methods of building such summaries are possible, but simple approaches such as the "tldr" approach mentioned above tend to work as well as more sophisticated alternatives, as we show in Section 5. Additional considerations in designing these prompts are elaborated in Appendix A.2 and A.3.

**(Weighted) Cluster Sampling.** Since the context size of existing LLMs is still limited, we cannot in general fit the entire dataset into the context for summarization. Furthermore, boosting algorithms require that we provide weak learners on *weighted* samples of the training set, effectively guiding the boosting process to focus on "harder" examples as the boosting process continues. Thus, instead of attempting to summarize the entire dataset, we propose to use only a representative subset of the dataset. The size of this subset is governed by the maximum context size and size of the data descriptions. To select this representative subset, we use weighted stratified sampling using subpopulations defined by clusters of language embeddings of each data description. The language embeddings are sentence representations generated by GPT-3. In particular, we use *hierarchical agglomerative clustering* [Nielsen, 2016] to identify clusters in the embedding. This process is shown in Algorithm 1. As we will show in Section 5, this process is able to consistently produce weak

---
**Algorithm 1** Cluster Sampling
---

1: **Input**: X, all training data; y, all training label; r, ratio of classes; p, AdaBoost weights of the current round; s, target number of samples.    ▷ r[k] is the proportion of examples in class k.
2: S ← new empty set
3: w ← new array with same length as X filled with -1. ▷ w[i] is probability of sampling example $i$.
4: **for** k = 1 to number of target classes in y **do**
5:     E ← **GPTEmbedding**(X[y == k])    ▷ E refers to the embeddings of the data descriptions
6:     C ← **AgglomerativeClustering**(E).    ▷ $C_j$ is set of data indices present in the $j^{th}$ cluster.
7:     c ← new empty array same size as $C$.    ▷ c[j] will store sampling probability of cluster $j$.
8:     **for** j = 1 to len(C) **do**
9:         c[j] ← $\frac{\text{len(X)}}{\text{len}(C_j)}$
10:     **end for**
11:     **for** i = 1 to len(X) **do**
12:         w[i] ← c[j], such that, i ∈ $C_j$
13:     **end for**
14:     w ← **Normalize**(**Normalize**(w) × p) ▷ **Normalize** turns weights to a probability distribution.
15:     Sample s × r[c] examples from X using categorical distribution w and append to S.
16: **end for**
17: **Return** S

---

---
**Algorithm 2** Summary Boosting (compact version)
---

1: **Input**: X, all training data; y, all training label; T: maximum number of rounds; s: size of the sampling subset; r: ratio of classes.    ▷ r[k] denotes the proportion of examples in class k.
2: h, $\epsilon$, $\alpha$ ← new empty arrays of length T.
3: N ← len(X); K ← number of target classes in y.
4: w ← new array of length N filled with $\frac{1}{N}$.                    ▷ w is the data distribution
5: **for** r = 1 to T **do**
6:     $(X_s, y_s)$ ← **ClusterSampling**(X, y, r, w, s) ▷ sample $s$ training examples from distribution w.
7:     h[r] ← **Summary**$(X_s, y_s)$                ▷ h[r] is the weak learner in the current round r.
8:     $\epsilon$[r] ← $\frac{\sum_{i=1}^{N} \text{w[i]} \times \mathbb{1}\{\text{h[r](X[i])} \neq \text{y[i]}\}}{\sum_{i=1}^{N} \text{w[i]}}$                ▷ $\epsilon[r]$ is the weighted error at round r.
9:     **if** $\epsilon$[r] ≥ $1 - \frac{1}{K}$ **then**
10:         Goto Step 6.
11:     **end if**
12:     $\alpha$[r] ← $\log\left(\frac{1-\epsilon[r]}{\epsilon[r]}\right) + \log(K-1)$ ▷ $\alpha$[r] refers to coefficient of the hypothesis at round r.
13:     **for** i = 1 to N **do**
14:         w[i] = w[i] × $\exp(\alpha[r]\mathbb{1}\{\text{h[r](X[i])} \neq \text{y[i]}\})$
15:     **end for**
16:     w ← **Normalize**(w)
17: **end for**
18: **Return** h, $\alpha$

---

learners, and able to improve upon random guessing under the distribution of interest (denoted by the input p to the algorithm). We share more details in Appendix A.7.

### 3.3  Boosting

Finally, we use the AdaBoost [Freund and Schapire, 1997] algorithm to produce an ensemble with these collections of summary-based weak learners. The central idea of AdaBoost is to fit a sequence of weak learners on repeatedly modified versions of the data. The algorithm is carried out over $T$ rounds, where the weights of the training data points are adjusted based on the training error.

Given a new data point, the predictions from classifiers from all rounds are then combined through a weighted majority vote to produce the final prediction. We use the error on a holdout validation set to determine the number of rounds T. A compact version of this process is presented in Algorithm 2. In the algorithm, the **Summary** method summarizes the examples in the prompt via the process discussed in Section 3.2. Each summary can be treated as a hypothesis that can classify new data.

However, unlike the summary process in Section 3.2, where we resample multiple times to find the best learner, the boosting process returns immediately when a summary with an error rate better than random guessing is found (refer Appendix A.2). We use **ClusterSampling** to subsample a mini-batch of examples that fit within the LLM's allowed context length. In Appendix A.10 and A.11, we provide a time complexity analysis of our method, including cost estimations for using APIs. Appendix A.8 covers the full version of our boosting procedure that works in practice.

## 4 Experiments

We conduct all of our experiments with OpenAI's GPT-3 API [Brown et al., 2020] and choose a collection of 18 tabular datasets from the UCI dataset [Dua and Graff, 2017] and OpenML [Vanschoren et al., 2014]. All main experiments are done with the `Curie` variant of GPT-3 unless otherwise specified, which has 13B parameters[1]. We compare the following methods:

- `Zero-shot`: query the language model with the data description and ask the model to complete the answer (refer Appendix A.4).

- `Few-shot`: provide a few labeled data descriptions of the training data as the context and ask the model to complete the answer for a new data description. To preserve consistency, we standardize the number of fewshot examples to approximately 15 for all datasets. The setting is explained in Appendix A.5.

- `Summary` (ours): generate a population of summaries given a list of data descriptions with cluster sampling and pick the summary with the lowest validation error; use the best summary as the context and ask the model to complete the answer for a new data description.

- `Summary Boosting` (ours): use `Summary` as a subroutine in AdaBoost.

Furthermore, we compared `Summary Boosting` against popular baselines for tabular data that do not use prompting:

- `KNN`: first embed the data descriptions with the GPT-3 embedding API [2] and then use K-nearest neighbor to classify a new data description. This simple baseline demonstrates how much information can the naive representation produced by LLMs provide about the tasks.

- `LIFT` [Dinh et al., 2022]: Language-Interfaced Fine-Tuning (LIFT) finetunes the LM with data descriptions (without binning) and their corresponding labels in a zero-shot.

- `TabPFN` [Hollmann et al., 2022]: TabPFN is a transformer-based architecture that performs Bayesian inference on the entire training and test data points at the same time.

- `XGBoost` [Chen and Guestrin, 2016]: XGBoost (eXtreme Gradient Boosting) is a regularized gradient boosting algorithm that is widely used for tabular data.

For each method and dataset, we use a $50/10/40$ split for train, validation, and test sets and repeat each experiment for 3 random seeds The results are shown in Table 1 and 2.

### 4.1 Analysis of prompting-based methods

As a general trend from Table 1, test performance improves in the order of `Zero-shot` < `Few-shot` < `Summary` < `Summary Boosting`. Firstly, unlike most works on zero-shot reasoning with LLMs, the LLMs do not have enough prior knowledge to make the correct prediction without additional information. As a result, we observe that `Zero-shot` performs poorly on all of the datasets. This observation highlights the necessity of learning from the data, and unlike other tasks, the LLMs themselves do not have enough built-in knowledge to succeed at tabular data zero-shot. Since `zero-shot` does not have enough prior knowledge to classify tabular data, we use few-shot in-context learning (`Few-shot`) to see if the added information helps make better predictions. As expected, on all the datasets other than `visualizing-hamster`, and `wholesale-customers`, `Few-shot` consistently improves the test performance compared to `Zero-shot`, suggesting that this added information is crucial for LLMs to work on tabular datasets.

---

[1]We use `Curie` because it is more cost-effective for large-scale experiments.
[2]https://beta.openai.com/docs/guides/embeddings

Table 1: Test errors for the chosen methods on all datasets (↓). **Data Type** indicates the number and types of attributes the dataset has (c is continuous and d is discrete). **Size** denotes the number of data points. In square brackets (if present) next to every dataset name, we provide its acronym referred to in our main text. In the bracket next to each dataset name is either the OpenML ID of the dataset or a reference to the dataset's associated publication. Error represents one standard deviation.

| Dataset | Data Type | Size | Zero-shot | Few-shot | Summary | Summary Boosting |
|---|---|---|---|---|---|---|
| caesarian [cae] (42901) | 1c4d | 80 | 0.425± 0.04 | 0.388± 0.02 | 0.350± 0.04 | **0.300± 0.04** |
| iris (61) | 4c0d | 150 | 0.680± 0.02 | 0.460± 0.01 | 0.275± 0.07 | **0.193± 0.03** |
| tae (48) | 1c4d | 151 | 0.556± 0.07 | 0.494± 0.01 | 0.474± 0.02 | **0.454± 0.03** |
| glass (41) | 9c0d | 214 | 0.486± 0.01 | 0.473± 0.01 | 0.466± 0.02 | **0.370± 0.02** |
| breast-cancer [bc] (13) | 7c5d | 277 | 0.754± 0.02 | 0.516± 0.02 | 0.337± 0.02 | **0.288± 0.02** |
| visualizing-environmental [ve] (678) | 3c0d | 111 | 0.522± 0.01 | 0.366± 0.01 | 0.304± 0.02 | **0.268± 0.03** |
| analcatdata-chlamydia [ac] (535) | 2c2d | 100 | 0.200± 0.00 | 0.200± 0.00 | **0.170± 0.01** | **0.170± 0.01** |
| wine (43571) | 13c0d | 178 | 0.820± 0.03 | 0.674± 0.02 | 0.475± 0.01 | **0.320± 0.01** |
| blood-transfusion-center [btc] (1464) | 4c0d | 748 | 0.544± 0.01 | 0.430± 0.00 | 0.258± 0.00 | **0.240± 0.04** |
| somerville-happiness-survey [shs] [Koczkodaj, 2018] | 0c7d | 143 | 0.416± 0.03 | 0.385± 0.03 | 0.422± 0.02 | **0.350± 0.02** |
| vehicle (54) | 18c0d | 846 | 0.765± 0.00 | 0.560± 0.01 | 0.510± 0.02 | **0.410± 0.04** |
| statlog-heart [stath] [Dua and Graff, 2017] | 6c7d | 270 | 0.551± 0.01 | 0.528± 0.01 | 0.444± 0.05 | **0.430± 0.01** |
| verterbra-column [vc] (1524) | 6c0d | 310 | 0.714± 0.03 | 0.435± 0.06 | 0.327± 0.01 | **0.262± 0.01** |
| ecoli (1011) | 7c0d | 336 | 0.581± 0.02 | 0.562± 0.01 | 0.480± 0.01 | **0.270± 0.03** |
| haberman-survival [hs] (43) | 3c0d | 306 | 0.308± 0.02 | 0.262± 0.01 | 0.277± 0.01 | **0.250± 0.01** |
| diabetes [dia] (37) | 8c0d | 768 | 0.446± 0.04 | 0.400± 0.00 | 0.360± 0.01 | **0.344± 0.01** |
| visualizing-hamster [hams] (708) | 5c0d | 73 | 0.464± 0.03 | 0.481± 0.05 | 0.360± 0.02 | **0.207± 0.00** |
| wholesale-customers [wc] (1511) | 6c1d | 440 | 0.364± 0.01 | 0.347± 0.01 | 0.349± 0.02 | **0.330± 0.00** |

Table 2: Test errors for chosen methods on all datasets (↓). (acronym and notation in Table 1)

| Dataset | Data Type | Size | Summary Boosting | LIFT | KNN | TabPFN | Xgboost |
|---|---|---|---|---|---|---|---|
| cae (42901) | 1c4d | 80 | **0.300± 0.04** | 0.312± 0.02 | **0.300± 0.00** | 0.425± 0.07 | 0.412± 0.05 |
| iris (61) | 4c0d | 150 | 0.193± 0.03 | 0.100± 0.01 | 0.106± 0.02 | **0.027± 0.00** | 0.054± 0.04 |
| tae (48) | 1c4d | 151 | 0.454± 0.03 | 0.480± 0.04 | 0.532± 0.01 | **0.450± 0.13** | 0.464± 0.01 |
| glass (41) | 9c0d | 214 | 0.370± 0.02 | 0.218± 0.02 | 0.294± 0.03 | **0.158± 0.05** | 0.254± 0.05 |
| bc (13) | 7c5d | 277 | 0.288± 0.02 | 0.318± 0.01 | 0.277± 0.02 | **0.264± 0.01** | 0.270± 0.01 |
| ve (678) | 3c0d | 111 | **0.268± 0.03** | 0.430± 0.04 | 0.308± 0.01 | 0.370± 0.04 | 0.279± 0.02 |
| ac (535) | 2c2d | 100 | 0.170± 0.01 | 0.180± 0.06 | 0.170± 0.01 | **0.090± 0.01** | 0.110± 0.04 |
| wine (43571) | 13c0d | 178 | 0.320± 0.01 | 0.065± 0.01 | 0.214± 0.05 | **0.040± 0.01** | **0.040± 0.01** |
| btc (1464) | 4c0d | 748 | 0.240± 0.04 | 0.270± 0.01 | 0.238± 0.00 | **0.209± 0.01** | 0.219± 0.01 |
| shs [Koczkodaj, 2018] | 0c7d | 143 | 0.350± 0.02 | 0.419± 0.02 | **0.326± 0.03** | 0.392± 0.00 | 0.406± 0.00 |
| vehicle (54) | 18c0d | 846 | 0.410± 0.04 | **0.111± 0.16** | 0.636± 0.01 | 0.178± 0.01 | 0.260± 0.00 |
| stath [Dua and Graff, 2017] | 6c7d | 270 | 0.430± 0.01 | **0.122± 0.17** | 0.244± 0.03 | 0.148± 0.03 | 0.215± 0.00 |
| vc (1524) | 6c0d | 310 | 0.262± 0.01 | 0.192± 0.04 | 0.318± 0.02 | **0.135± 0.00** | 0.187± 0.04 |
| ecoli (1011) | 7c0d | 336 | 0.270± 0.03 | 0.126± 0.03 | 0.211± 0.03 | **0.036± 0.02** | 0.066± 0.01 |
| hs (43) | 3c0d | 306 | **0.250± 0.01** | 0.314± 0.03 | 0.278± 0.00 | 0.262± 0.02 | 0.281± 0.02 |
| dia (37) | 8c0d | 768 | 0.344± 0.01 | 0.324± 0.01 | 0.353± 0.02 | 0.238± 0.03 | **0.234± 0.00** |
| hams (708) | 5c0d | 73 | **0.207± 0.00** | 0.334± 0.08 | 0.528± 0.02 | 0.328± 0.01 | 0.411± 0.01 |
| wc (1511) | 6c1d | 440 | 0.330± 0.00 | 0.125± 0.04 | **0.043± 0.00** | 0.088± 0.00 | 0.098± 0.02 |

Unlike naively stacking examples inside the prompt as in Few-shot, Summary condenses knowledge from these examples and is the first important algorithmic component of our framework for creating weak learners using LLMs. We see that Summary consistently improves upon Few-shot on all the datasets other than haberman-survival and wholesale-customers. This observation suggests that summarization is a powerful way to improve few-shot performance and has the potential for even other tasks using LLMs. Finally, for every dataset we tested, boosting with summarization consistently outperforms all other prompting-based approaches. This observation corroborates our hypothesis that *LLMs with summarization are a good candidate for creating weak learners in boosting*.

## 4.2 Comparison to other tabular methods

In Table 2, we also observe that LLMs have a hard time reasoning about continuous attributes without finetuning, especially on the glass, wine, iris and vehicle datasets. In particular, when the datasets have many continuous features, the performance of Summary Boosting can be considerably worse than other methods such as LIFT or Xgboost. This may be because LLMs are fairly bad at quantitive reasoning without finetuning [Lewkowycz et al., 2022], which may be overcome in future LLMs.

While KNN is a relatively simple baseline, its performance is surprisingly good at a few tasks such as wholesale-customers, statlog-heart, ecoli and wine. This highlights that LLMs have a remarkable amount of general prior knowledge about the world compared to methods like XGboost that sometimes this knowledge alone can produce good performances.

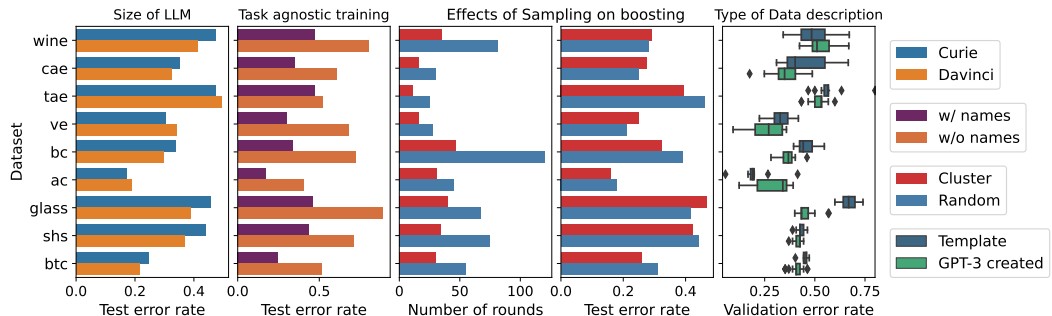

Figure 3: Ablation experiments compared for the `Summary` method. The dataset acronyms are referred to 1. **Left:** Performance of `curie` vs `davinci`. **Second from left**: Comparison with and without task-specific attribute names. **Middle**: Effect of Cluster vs. Random sampling on the number of rounds till convergence and **Second from Right**: their final test errors. **Right:** Performance of templatized vs LLM-generated data descriptions.

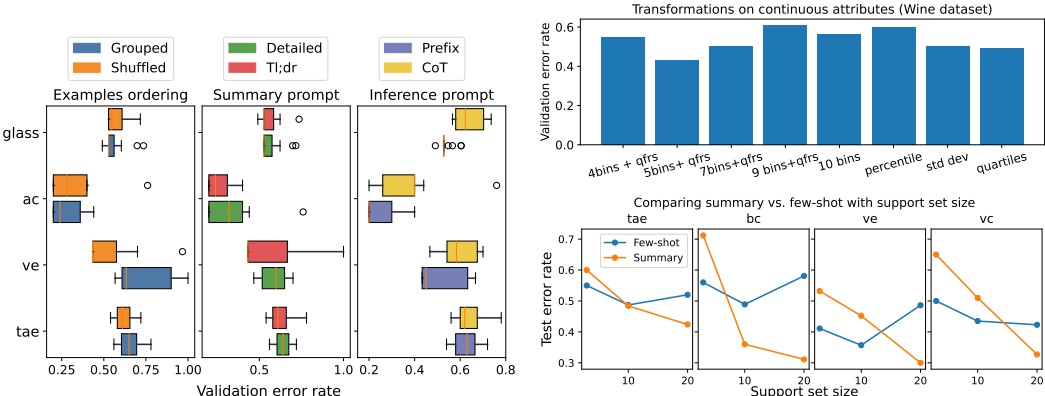

Figure 4: Additional ablations for the `Summary` method. **(Left)** Prompt design choices. The first plot shows the effect of shuffling examples vs presenting them by class. The center plot compares `tldr` vs a more explicit prompt for inducing summary. The last plot compares prompts for doing inference. **(Right Top)** Performance of methods for discretizing continuous attributes on the *Wine* dataset. **(Right Bottom)** Performance of `Few-shot` and `Summary` as a function of the number of examples in the context.

Finally, we observe that `Summary Boosting` performs very well when the size of the dataset is very small. This makes sense since the strength of using LLMs as weak learners is that they have a large amount of generic prior about the world from pre-training. When the dataset is large, this prior knowledge might become less relevant and methods like finetuning become more competitive.

## 5 Ablations

Summarization forms the core of our methodology for generating weak learners. Consequently, it becomes important to identify an ideal setting that can induce high-quality summaries. We perform ablation studies over the `Summary` method, to decide hyperparameters for getting a good weak learner.

**Preprocessing of continuous attributes.** We tried several encoding techniques for continuous features, including binning, percentiles, and standard deviations. We chose the approach of describing them in technical language terms as well as assigning quantifiers for each level, as illustrated in Figure 4 right top. We observed that binning with quantifiers such as "low", "medium", and "high" was most effective for comparing examples and generating high-quality summaries. After hyperparameter tuning, we identified that using 5 bins provides sufficient granularity to distinguish variations in the continuous values. More details can be found in the Appendix A.6.

**Does the LLM explore prior knowledge to infer?** To demonstrate the LLM's utilization of prior knowledge, we conduct an ablation study by masking the attribute names and using a template *"This example has features f1 = {}, f2 = {} and so on."* Figure 3 (second from left) shows the result. Using true variable names in the data descriptions leads to superior few-shot learning performance compared to using dummy names. This confirms that the model indeed leverages its prior knowledge of variables for predictions.

**How does model size affect the performance?** A natural question to ask is how the model size affects the downstream performance. We compare the `Summary` performances of GPT-3-`davinci` (175B parameters) and GPT-3-`curie` (13B parameters) on 5 datasets in Figure 3 (left). Surprisingly, we find that the larger model (`davinci`) does not consistently improve upon the smaller model. We also compare ChatGPT in A.12 and discuss the effects of RLHF [Ouyang et al., 2022]. We further show that our method generalizes to LLMs other than GPT, by comparing its performance with Claude-2 in Appendix A.13).

**How does the performance scale with more examples?** In Figure 4 (right bottom), we study how the behavior of `Few-shot` and `Summary` change with different support set sizes (i.e., the number of data descriptions that are summarized). `Few-shot` performance reaches an optimal size around the medium context length and degrades with more examples. In contrast, `Summary` improves with more data, which is the more desirable behavior.

**Ordering of examples.** Unlike conventional machine learning models, the ordering of examples in summarization affects the generation of hypotheses. There are two approaches: 1. presenting descriptions randomly (`shuffled`), and 2. grouping descriptions by classes (`grouped`). In Figure 4 (left), we compare these approaches on 4 datasets with `Summary` and find no significant difference. We use `shuffled` for all other experiments.

**Different summary and inference prompts.** The LLM easily generates concise summaries on standard datasets like `iris` using a simple "`tl;dr`" prompt, but requires a more explicit prompt on complex datasets like "vertebra-column". Comparing their performance in Figure 4 (left), both prompting modes are equally effective, so detailed prompts were used in all other experiments. See Appendix table 3 for the complete list of summary prompts used. In the left of Figure 4, we also compare two strategies for inference - prefix prompt (e.g. "This flower will be classified as"), and two-stage chain-of-thought prompting (e.g. "Let's think step by step") [Kojima et al., 2022]. We observe no statistically significant difference between them[3]. Since the prefix prompt more often completes the query well under lesser compute, we use prefix prompt for all the other experiments.

**Texts generated from template vs. GPT.** In Figure 3 (right), we see that using GPT-generated data descriptions consistently achieves better results. Refer to Appendix A.9 for examples of these templates. This may be due to the fact that the data description generated by LLMs conforms to natural text distribution more closely, which is desirable for performing inference using LLMs.

**Effect of cluster sampling.** Cluster sampling improves the performance of the LLM by selecting a representative set of texts that generalize better, reducing validation error faster compared to random sampling during boosting. Although it may require more resampling to achieve a weak learner, cluster sampling converges much faster than random sampling as we see in Figure 3 - middle and second from right. However, with sufficient boosting rounds, the performances of the two sampling methods are not statistically different.

## 6 Limitations

Although the weak learning approach we develop here shows promise, there are currently several drawbacks. Although summarization and boosting alleviate manual prompt tuning to a large extent, we still had to minimally tune some parts of the pipeline to get ideal performance (see Appendix table 3). Additionally, when the dataset contains many continuous attributes, there is a non-trivial gap between `Summary Boosting` and the other methods such as XGBoost or finetuning. Finally, the

---

[3]We do observe that CoT performs better with larger models.

max input length of GPT-3 makes it harder to generate good summaries with just subset sampling. Eventually on larger datasets, after a certain number of boosting rounds, the summaries derived from a subset of examples may not further decrease the weighted error across all training examples. Rigorous techniques such as structured prompting handle this issue by rescaling attention weights [Hao et al., 2022]. We believe this issue could be solved with more powerful LLMs such as GPT-4. Furthermore, methods like ours built using LLMs inherit bias that comes from the pretraining data. Specifically, it can affect the model's ability to objectively summarize the examples or make predictions that are inconsistent with the biased pre-training data. We believe with better and debiased LLMs these issues can be alleviated. We want to emphasize that our experiments utilized OpenAI APIs. In the appendix A.14, we provide a brief discussion on the impacts of commercial APIs.

## 7 Conclusion

LLMs have been widely used in recent years, not just for their generative abilities but for their ability to serve as zero- or few-shot learners with proper prompting. This work aims to situate them within another context of "simple" learning algorithms – the weak learner paradigm that forms the foundation of boosting approaches. We show that leveraging the summarization capabilities of LLMs indeed leads to models that function as weak learners, and which can thus be integrated into boosting frameworks. "Summary" as an intermediate step makes reasoning easier and works inside boosting algorithms for small tabular data up to 300 data points without many continuous features. Overall, this result leads to new potential paradigms for treating the results of LLM prompting not just as individual predictors, but as part of a larger set of meta-models as well.

## Acknowledgements

Yiding is supported by funding from the Bosch Center of Artificial Intelligence. We would like to thank the anonymous reviewers for their valuable feedbacks.

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

# A  Failure Modes

The design of the prompt plays a pivotal role in our entire process. Specifying instructions precisely can create a significant difference, whether it comes to effectively describing tabular data, generating reliable summaries or inferring accurate predictions (shown in Figures 1 and 2). While soft prompting has been successful at instructing LLMs [Lester et al., 2021], it cannot be applied to our setting because our classification algorithm learns a human-level summary as a prompt for classifying data, rather than a soft prompt. Instead we chose to write prompts that ask the LLM to perform these tasks. In this way, our prompting method is entirely gradient-free. Hand-written prompts also offer flexibility, aligning well with our core methodology for creating weak learner. While carefully handcrafting prompts this way might seem intensive, we will show that once we identify the ideal hyperparameter settings they can be framed with little effort.

## A.1  Data Conversion Challenges

Language models (LLMs) have demonstrated impressive performance on standard tasks with minimal supervision [Wang et al., 2022b, Brown et al., 2020]. However, for converting tabular data to text, there were several considerations to meet the requirements of our task. As highlighted in Section 3.1, we will refer to these texts as *data descriptions*.

**Ensuring Uniform Length**  Firstly, the data descriptions should not be too long or short, also be of comparable length. Excessively long descriptions limit the number of examples that can be fit inside the prompt and summarized. We observed in Figure 4 (right bottom) that the summary performance also scales with more examples, so it makes sense to accomodate as many of them as possible by having descriptions of approximately uniform length.

A straightforward way of achieving this uniformity would be by specifying a max word length as part of the conversion prompt itself, as in "Describe in not more than 80 words". However, we found this approach can falter, sometimes leading to overly simplistic descriptions like "These are annual spendings of a customer." (in the `wholesale-customers` dataset).

Consequently, we adopt more nuanced strategy by first modifying the prompt with the terms "concisely" and "accurately" to emphasize the brevity and preciseness of the generated descriptions (shown in Figure 1). Then, we implement a resampling strategy, that generates descriptions until finding the one with a desired length ranging between 20 to 80 words. This process achieves consistent and uniformly long descriptions.

**Including Metadata**  Prepending metadata to the prompt enhances the contextual awareness of the task, resulting in higher-quality descriptions (shown in Figure 1).

**Separating Features from Labels**  In our method, the data descriptions function dual role, both as training examples and as query for inferring class labels. This suggests that, when converting data to text, the features need to be described separately from the target label as illustrated in Figure 1. The resulting strings are then concatenated to form the data description. Instead, if the whole tabular record were passed to the LLM, it often produces texts that assimilate the classification label information in the meat of the description itself, rendering it difficult to extract a query for doing inference.

While one might cleverly come up with prompts that can allow the LLM to describe the features and target label in separate sentences, we found it to be more sensible to supply just the features for describing and not reveal any information about the target task. Sometimes that can liberate the LLM to hallucinate some facts about the task and form biased data to begin with.

**Natural-Sounding Descriptions**  While the LLM generates a different-styled response every time, to explicitly ensure that the generated descriptions are not template-like by chance, add a directive at the end of the prompt: "Use your creativity". This encourages the LLM to produce more natural narratives of the record. Alternatively, setting a higher temperature during decoding achieves a similar effect.

## A.2 Summarization

There are several aspects worth considering that can contribute to high-quality summaries.

**Sampling candidate summaries**  A well-crafted summary is a one that captures salient information of the dataset, in a way that facilitates inferring predictions off it. However, the process of generating summary using a LLM is inherently stochastic due to temperature sampling, as a result, the generated summary can be noisy. From our experiments with tuning this temperature, we found $0.80$ to be ideal through Bayesian optimization. Even at this value, on average only 1 out of 3 summaries were meaningful.

A noisy summary can be distinguished quite easily. For instance, on the `vehicle` dataset, the `tl;dr` prompt elicits summaries as naive as "The given data describes a bus, van, or saab silhouette." or "The data in this table identifies a vehicle as a bus, saab, opel, or van. The compactness, circularity, distance circularity, radius ratio, hollows ratio, and symmetry are all predictive of the vehicle's type." which does not offer actionable insight.

This observation indicates that summaries need to be sampled quite a few times and the best one can be determined based on the validation error. As a result, for the `Summary` learning procedure in Section 3.2, we resample approximately 25 times to find a good summary. Also, given that our datasets are small, it is not unusual for the summaries to have the same validation error. When tied, we pick one having a higher training error rate, i.e. lower generalization gap.

Differently, in our `Summary boosting` procedure explained in Section 3.3, we resample only until finding a summary whose training error is better than random guessing and return immediately.

**Ordering examples inside the *summarization* prompt**  Unlike gradient descent, prompting is not robust to the presentation of the examples to the learning algorithm. While we show via ablation studies in Section 5 that there is no statistically significant difference in performance between either shuffling the examples or listing them by class, we can generally expect that depending on the dataset and the number of target classes, one might be preferred over the other.

For instance, in a multi-class setting, listing examples by class might be more helpful in reaching a weak learner quickly. However, in a two-class setting, the summary might actually benefit from randomness in the shuffled examples.

**Customizing the *summarization* prompt**  The way of asking the LLM to summarize examples can also give rise to good/bad summaries. For instance, one can prompt the LLM with a simple `tl;dr` or specify the task more elaborately. We will refer to the latter option as `explicit`. As we demonstrated in Figure 4 (left), both are means to the goal and do not statistically differ in terms of performance induced.

However, in our experiments on certain datasets, we would rather be incentivized choosing the `explicit` over the `tl;dr` to attain a weak learner more quickly. This choice becomes important purely for compute reasons as it will take relatively lesser resampling, while the `tl;dr` still works. For instance, this scenario can happen when the LLM cannot decipher what the summary is supposed to say, by just observing the examples. As examples, the `tl;dr` prompt suffices on datasets such as `iris`, `diabetes`, and `wine` that are commonly encountered in prediction context, whereas the LLM might not be very familar with the goals of `vertebra-column` or `somerville-happiness-survey` data, necessitating the use of the `explicit` prompt. For these other datasets, the fact that it is a classification problem based on some features and target classes may not be very apparent from just the examples and metadata. So, providing a directive such as "Summarize in detail how we can tell apart people with normal and abnormal vertebra-column" reduces ambiguity in the task setup and reduces probability of a noisy summary.

While manual intervention is necessary, framing this prompt can be done with little effort. We provide a comprehensive list of these parameters for all datasets in Table 3.

**Including Metadata**  Similar to data conversion, including meta-data information in the prompt offers better penetration into the world of the dataset, as a result improves boosting performance.

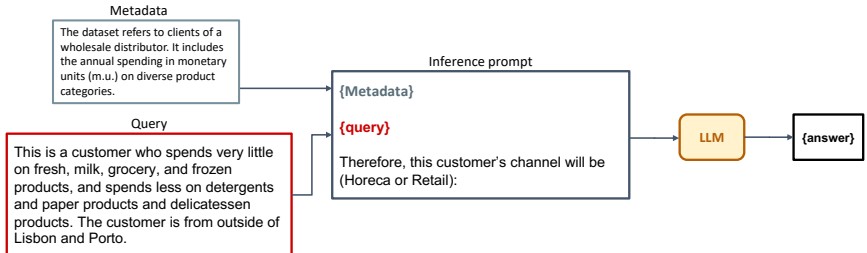

Figure 5: Steps of Zeroshot prompting

## A.3 Inference

**Mapping labels from LLM responses**  Answer mapping refers to the process of assigning the model's answer to a target output class. This step might be trivial when the answer is the class itself, for example when the LLM responds with "non-relapse" or "recurrence" to a query on the `breast-cancer` dataset. However, in other instances, it can become tricky when the LLM's responses are "will not recur" or "has higher chance of non-relapse than recurrence", requiring a more complex decoding logic to identify the target class.

Previous works have handled this problem by disguising the task as a Cloze prompt and learning a verbalizer, i.e. MLP projection of hidden state of the **[MASK]** token, that maps the predicted token to the target classes [Cui et al., 2022, Hu et al., 2021]. By training a verbalizer, one can determinsically go from token vocabulary to the label space. There also exist unsupervised statistical techniques for achieving label mapping [Wang et al., 2022a].

In our method however, we strictly interact with the LLM through prompts and do not access the hidden state nor gradients. As a result, our inference process shown in Figure 2 focusses on inferring the class label solely through prefix prompts, without relying on learning an explicit mapping. Specifically, by conditioning on a suitable prefix, we constrain the LLM to return exactly the class label string. For example, the prefix "Therefore this iris flower is likely to be (setosa, versicolor, virginica):" works for the `iris` dataset. A key observation guiding the design of such a prefix prompt is the fact that specifying the output classes entices the LLM to predict from among these classes. With a rather plain prefix like "Predict what will be the type of this flower.", the LLM's answer space is unconstrained and it might liberally go on to explain a chain of reasoning such as "The flower has short petals and long sepals, hence it is versicolor, and not setosa." preventing a simple keyword search for the class label.

For a full list of these inference prompts, refer Table 3.

**Two-step prompting, Davinci vs. Curie**  It is worth mentioning that a two-step prompting trick, by first calling "Lets think step by step" then concatenating the response with the prefix prompt also results in accurate answers, as we have shown in Figure 4 (left). However, it could only be implemented on the larger model `Davinci` but not `Curie` which is primarily used in our experiments. Interestingly `Davinci`'s chain of thought reasoning even outperforms its prefix prompt counterpart. In all our experiments with Curie however, the prefix technique works reasonably well.

The Davinci API also offers a suffix argument which can be invoked for predicting in a more natural way. For instance, for the `breast-cancer` dataset, the prompt can be posed with prefix "All in all, this woman is more likely to have a" and a suffix " of breast cancer." directly expecting the LLM to fill in with "recurrence" or "non-relapse."

## A.4 Zeroshot Setting

We extend the analysis of prompting-based methods in Section 4.1 by further delving into the `Zeroshot` experiment. We illustrate this experimental setting in Figure 5. It only consists of the inference prompt, wherein the LLM is presented with the metadata and a query. To facilitate answer mapping, the output classes are also indicated in the prompt. Unlike `Summary`, the `zeroshot` process

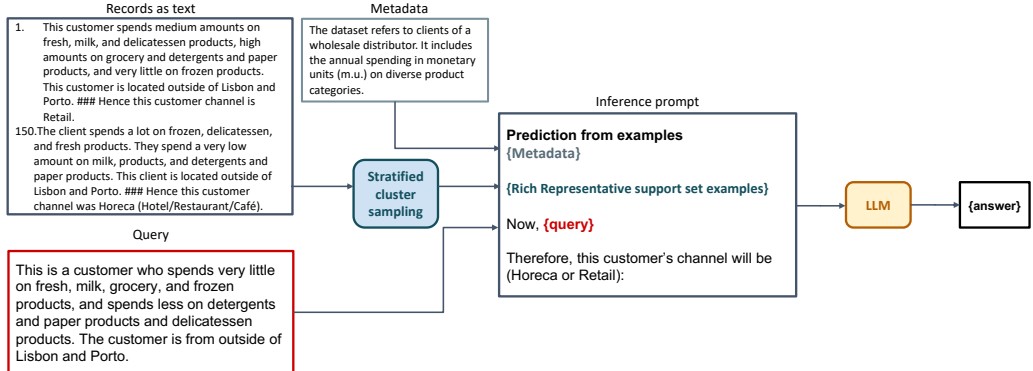

Figure 6: Workflow in fewshot prompting

is not stochastic as there is no learning involved. For inference, the predicted tokens are sampled greedily at temperature $= 0$.

### A.5 Few-shot Setting

Extending the analysis from Section 4, we explain the `Fewshot` setting. As illustrated in Figure 6, it is an instance of in-context learning, where the support set examples are listed inside the prompt attached with a query in the end. The support set was chosen from the training set through the stratified cluster sampling outlined in Algorithm 1. This results in a semantically diverse collection of examples evenly spread across the classes. Since we observe in Figure 4 (right bottom) that the `Fewshot` performance drops with more examples, we choose approximately 15 examples to fit in the prompt.

Similar to the `Summary` method, this inference prompt carries meta-information about the dataset, and also indicates the output classes. The prompt also injects support set examples that are stringed together as we show in Figure 6. The predictions are made greedily at a temperature of 0.

Again, the `Fewshot` method is a stochastic process whose outcome depends on the selection of the support set. So finding the ideal prompt, requires sampling different support sets from the training set quite a few times. We perform this resampling approximately 25 times and pick the best prompt based on the validation error.

### A.6 Preprocessing continuous attributes

Extending from the ablation studies in Section 5, we demonstrate in Table 4, concrete examples of these encoding techniques applied to continuous features. Every numerical column in the dataset was subject to the transformation independently of the rest.

We applied several encoding techniques for continuous features, including binning, percentiles, and standard deviations. Our approach involved using technical language terms to describe these ranges, such as *"falls in the nth bin/nth percentile or n deviations above/below mean"*. We also characterize them in a more naturalistic way by assigning quantifiers such as *low*, *medium*, and *high* to each level in the binning technique.

To create effective textual descriptions, we examined three high-level approaches: 1. presenting only numerical values, 2. using solely textual encodings, and 3. concatenating both. We observed that utilizing textual encoding alone outperformed the other methods. As a result, we focused on mainly comparing textual encoding methods as shown in Figure 4 (right top). Through Bayesian optimization, we found that binning with "5" quantifiers was ideal for generating high-quality summaries.

We describe each encoding technique as follows:

- **Binning**: It involves creating a histogram with the given number of bins. As outputs, the values are directly described as *"falling in the n-th bin"* as illustrated in the `10 bins` experiment. However, in the presence of *degree quantifiers* which are categorical names

assigned to the bins, these tags are used instead. We found that as opposed to calling out the bin number, describing in terms of these quantifiers further aids the LLM in comparing the relative extent to which features match and improving the estimation of similarities. This led us to tune the number of bins against these degree quantifiers, selecting values in the range of 4, 5, 7, and 9 bins. The first four rows in Table 4 show how these tags get translated into the record.

- **Percentile**: It is given by computing the percentile rank of a value relative to that series of values. Then, the value is described as falling in that percentile rank in words. This is closer to representation of the true numerical values per se, but helps the LLM draw comparisons on a scale of 1-100.

- **Standard deviations**: In this procedure, the values are segmented into six ranges based on distance from the mean, given by one/two/three standard deviations above/below the mean.

- **Quartiles**: Here, we consider the first and third quartiles, and the median as landmarks to bucketize the values into four partitions.

Among these methods, the "5 bins with quantifiers" strikes a balance in granularity scale. It is not excessively fine-grained as "percentile", nor overly abstract, as the "4-bin" approach. This balance ultimately leads to optimal performance.

## A.7 Cluster Sampling components

We discuss more of the functions in Algorithm 1.

**GPT-Embedding** is OpenAI's text similarity model `text-embedding-ada-002` that takes a maximum input size of 8191 tokens. It returns a 1536-dimensional embedding for text. OpenAI recommends cosine distance for comparing *ada* embeddings in downstream tasks.

As a result, the **AgglomerativeClustering** algorithm applies hierarchical clustering over these features using cosine distance, average linkage and a heuristically selected distance threshold of 0.05. It yields a set of clusters $C$ and each $C_j$ contains a list of indices of data points that belong to that cluster $j$.

## A.8 Adaboost Optimizations

We additionally apply several run-time optimizations to the boosting algorithm described in 3.3. Thus we present its full version in Algorithm 3.

- **Raising the bar for a weak learner:** Our goal was to create high-quality summaries that dramatically reduce the validation error rate and significantly accelerate the convergence of the boosting procedure. Thus we raise the performance threshold to a notch slightly higher than random guessing probability (see Step 15 in Algorithm 3), provoking insightful summaries.

  We resample until finding a weak learner that satisfies this threshold.

  The positive quantity $\mu$ is a hyperparameter that typically takes values 0.08 for 2-class problem and 0.16 for 3-class problem, and so on.

  Although this step increases compute, it yields better weak learners and improves convergence overall.

- **Permuting the predicted class label assignments:**

  We harness the potential of permuting the class assignments by exploring $K!$ different mappings of predictions to classes using the **PermutedLabelMappings** function in steps 11-14. This process helps us identify the mapping that minimizes the training error to the greatest extent.

  By considering multiple permutations of predictions across the label space, as outlined in Steps 11-14 of Algorithm 3, we obtain a hashmap $p$ from the **PermutedLabelMappings** function. This hashmap maps the predictions $\hat{y}$ to the permuted label space. Selecting the mapping, $p^*$ that results in the lowest training error effectively diminishes the cumulative training error during boosting iterations and proves to be an effective strategy for generating strong weak learners. This technique is particularly advantageous in scenarios involving more than two classes.

---
**Algorithm 3** Summary Boosting
---
1: **Input**: $X$, all training data; $y$, all training label; $\mathtt{T}$: maximum number of rounds; $\mathtt{s}$: size of the sampled subset.
2: $\mathtt{h}, \mathtt{P}, \epsilon, \alpha \leftarrow$ empty array of size $\mathtt{T}$. $\triangleright$ $\mathtt{h}$ holds the round-wise hypotheses, $\mathtt{P}$ are the corresponding label mappings, $\epsilon$ gathers the weighted train errors, and $\alpha$ are coefficients of the hypotheses.
3: $\mathtt{N} \leftarrow \mathtt{len}(X)$
4: $\mathtt{c} \leftarrow$ set of target classes
5: $\mathtt{K} \leftarrow \mathtt{len}(\mathtt{c})$
6: $\mathtt{w} \leftarrow$ new array of size $\mathtt{N}$ filled with $\frac{1}{\mathtt{N}}$. $\qquad\qquad\qquad$ $\triangleright$ $\mathtt{w}$ is the weighted data distribution
7: **for** $\mathtt{r} = 1$ to $\mathtt{T}$ **do**
8: $\quad$ $(X_s, y_s) \leftarrow$ Cluster-sample $s$ examples from training distribution $\mathtt{w}$.
9: $\quad$ $\mathtt{h}[\mathtt{r}] \leftarrow$ **Summary** $(X_s, y_s)$ $\qquad\qquad$ $\triangleright$ $\mathtt{h}[\mathtt{r}]$ is the weak learner in the current round
10: $\quad$ $\hat{y}[i] \leftarrow h[r](X[i])$ $\qquad\qquad\qquad$ $\triangleright$ $\hat{y}$ refers to predictions on training set
11: $\quad$ $\xi \leftarrow$ empty hashmap $\qquad$ $\triangleright$ $\xi[p]$ will have error rate of the corresponding label mapping $p$
12: $\quad$ **for** $p$ in **PermutedLabelMappings**($\mathtt{c}$) **do**
13: $\qquad$ $\xi[p] \leftarrow \frac{\sum_{i=1}^{N} \mathtt{w}[i] \times \mathbb{1}\{p[\hat{y}[i]] \neq y[i]\}}{\sum_{i=1}^{N} \mathtt{w}[i]}$
14: $\quad$ **end for**
15: $\quad$ $p^* \leftarrow \arg\min_p \xi[p]$
16: $\quad$ **if** $\xi[p^*] > 1 - \frac{1}{K} - \mu$ OR **AllSame**($\hat{y}$) **then** Goto Step 8.
17: $\quad$ **else**
18: $\qquad$ $\mathtt{P}[\mathtt{r}] \leftarrow p^*$; $\epsilon[\mathtt{r}] \leftarrow \xi[p^*]$
19: $\quad$ **end if**
20: $\quad$ **if** $\epsilon[\mathtt{r}] == 0$ **then**
21: $\qquad$ Break
22: $\quad$ **end if**
23: $\quad$ $\alpha[\mathtt{r}] \leftarrow \log\left(\frac{1 - \epsilon[\mathtt{r}]}{\epsilon[\mathtt{r}]}\right) + \log(\mathtt{K} - 1)$
24: $\quad$ **for** $i = 1$ to $\mathtt{N}$ **do**
25: $\qquad$ $\mathtt{w}[i] = \mathtt{w}[i] \times \exp(\alpha[\mathtt{r}] \mathbb{1}\{\mathtt{P}[\mathtt{r}][h[\mathtt{r}](X[i])] \neq y[i]\})$
26: $\quad$ **end for**
27: $\quad$ $\mathtt{w} \leftarrow$ **Normalize**($\mathtt{w}$)
28: **end for**
29: **Return** $\mathtt{h}, \alpha$
---

- **Sanity checks:** Finally, to ensure robustness of the weak learner when faced with skewed datasets, we have implemented a policy that disallows a naive all-ones classifier. The condition calling **AllSame** in Step 15 of Algorithm 3 performs this check.

## A.9 Text Templates

In the ablation study which involves comparing the descriptions created manually vs. by a LLM, as illustrated in Figure 3 (right), we transform the descriptive attributes into the textual format by applying a pre-defined template. In Table 5 we provide examples of these templates for selected datasets.

## A.10 Complexity Analysis

We provide the time complexity analysis comparing our boosting procedure to finetuning the LLM.

For finetuning, the complexity is $\mathcal{O}(TNf)$, where $f$ is runtime of the LLM, $T$ is number of epochs, $N$ is the number of data points.

For summary boosting, the complexity is $\mathcal{O}(TRf)$, where $f$ is runtime of the LLM, $T$ is number of boosting rounds and $R$ is the number of resampling per round.

Concretely, for a dataset with 175 examples, finetuning takes 20 epochs $\times$ 175 examples $\times$ 2 = 7000 passes through the LLM. 2 stands for both forward and backward passes through the model.

For the same dataset boosting requires 50 rounds $\times$ 25 resampling on average = 1250 passes through the LLM.

Thus, we believe the complexity of our algorithm is at least comparable to, if not better than, that of finetuning (without considering the cost of the actual API calls).

## A.11 Estimating the cost of API calls

While our method is applicable to any large language model, we primarily conducted experiments using GPT-3. Each API call to GPT-3 incurs a specific dollar cost.

After analyzing the running time complexity of summary boosting, which is $\mathcal{O}(TRf)$, we can provide a rough estimation of the cost associated with training a classifier on any given dataset.

To begin, when making a call to summarize examples, the prompt is filled up to the maximum context length, which is 2048 tokens for the query prompt and completion. We'll refer to these summary tokens as $S_t = 2048$.

Additionally, if $N$ represents the size of the dataset and we allocate $(50 + 10)$

Now, to obtain a weak learner at boosting round $r$, we may need to resample up to $R$ candidate summaries. Furthermore, we calculate the training error for each candidate summary to determine if it performs better than random guessing. Once the desired weak learner is found, we compute the validation error for that round only once. Therefore, each round requires querying $R \times (S_t + 0.5N \times P_t) + 0.1N \times P_t$ tokens.

Considering that the maximum number of rounds is denoted as $T$, the total number of tokens exchanged would be $T \times [R \times (S_t + 0.5N \times P_t) + 0.1N \times P_t]$.

For instance, let's consider a dataset with 175 examples. In this case, the cost would be 30 rounds $\times$ [20 resampling $\times$ (2048 summary tokens + (0.5 $\times$ 175 training examples) $\times$ 210 prediction tokens) + (0.1 $\times$ 175 validation examples) $\times$ 210 prediction tokens] = 12364050 tokens, which approximately costs \$25 for Curie at a rate of \$0.002/1K tokens.

## A.12 Can ChatGPT function as a weak learner?

One would expect that it is more advantageous to try newer LLMs such as ChatGPT that produce increasingly more human-like text and are far more sample-efficient, i.e. can summarize more examples since they come with a larger context length. To investigate this, we conduct experiments by feeding ChatGPT with the same tabular data descriptions and using identical prompts to create weak learners. The results are presented in Table 6.

Surprisingly ChatGPT outperforms Curie in classifying datasets with more numerical features, such as `wine`, `wholesale-customers`, and `iris`. This observation suggests that LLMs are becoming more adept at quantitative reasoning from finetuning with more data. However, the reinforcement learning from human feedback (RLHF) [Ouyang et al., 2022] poses a limitation as it still ensures the generated text does not deviate too much from its prior. The generated text distribution adheres closely to the behavior programmed into the LLM induced by optimizing with such a reward model. Consequently it becomes challenging to bias the LLM with adversarial examples that might occasionally emerge in the training set.

For example, ChatGPT does not mostly generalize well on datasets with medical information such as `verterbra-column`, `breast-cancer`, `caesarian` and `blood-transfusion-center` where there can be examples contrary to common medical beliefs. In these cases, the RLHF is more restrictive due to its conformity to human preferences and does not neutrally summarize examples at hand from a classification standpoint. However, boosting imposes a significantly higher penalty on examples that the model fails to classify correctly, causing ChatGPT to not decrease training error after a few epochs. While these models exhibit promise in terms of higher-order problem-solving skills, their capabilities can also be limited by their alignment with human preferences.

### A.13 Others LLM providers: Claude-2

Summarizing documents and text-based reasoning are standard use cases of LLM, so in principle we expect `Summary Boosting` to work with other prompt-based LLMs. To validate this claim, we tested our method on other LLMs different from GPT, specifically Claude-2. The results for four datasets are given in Table 7. We see that the performance of Claude-2 is slightly worse than that of GPT-3 Curie (but better than ChatGPT). This experiment demonstrates that our method still works outside GPT-3, and is able to produce weak learners for different base LLMs. Though the scope of the experiments is limited, it shows preliminary evidence that the method is robust across models. On the other hand, Claude-2 released by Anthropic is accessed through a commercial API. In section A.14 we discuss the limitations of using commercial APIs for summary boosting.

### A.14 Implications of using commercial API for LLM experiments

Our experiments are done using OpenAI's API access. We chose to use APIs instead of hosting the LLMs because it lets us test the capabilities of the models at a reasonable efficiency as boosting requires many sequential calls to the language model and we need to do it on many datasets in parallel. At the current pace of innovation in LLMs, we are cautiously optimistic that there will be software solutions (e.g., for efficient inference) that will change the circumstances. In this section, we discuss the limitations and effects of using LLMs for ML research in general.

- **Limited Customization:** Commercial APIs may not allow fine-tuning the model according to our specific needs. We might not have control over the architecture, hyperparameters, or training data used for the model, which can be crucial for certain research tasks. The backend models offered by these vendors are also continuously updated either through new training methods or more data. For ML research in general, it will be a good practice to mention the version of the model for replicating the experiments.

- **Lack of Transparency:** The inner workings and architecture details of commercial APIs are often proprietary and not fully disclosed. This lack of transparency can make it challenging to understand model behavior, biases, and limitations. Specifically, it deprives us of the opportunity to study the training process itself, which can be valuable for understanding how these models learn and for improving our own research methodologies. Commercial APIs may inherit biases present in the training data, potentially leading to biased or unfair responses.

- **Data Privacy and Security:** Using a commercial API involves sending the data to an external server, which might raise concerns about data privacy and security, especially if we are dealing with sensitive or confidential information. There might also be regulatory and legal considerations when passing sensitive data to these API.

Table 3: **Prompt design:** Prompt parameter settings for every dataset.

| Dataset | Prompting hyperparameters |
|---------|---------------------------|
| caesarian | *metadata:* This dataset contains information about caesarian section results of 80 pregnant women with the most important characteristics of delivery problems in the medical field.The goal is to predict whether a woman will undergo normal or caesarian delivery.
*classes:* [normal, caesarian]
*summary directive:* Tl;dr
*inference directive:* Hence this woman's delivery mode is likely to be (normal or caesarian): |
| iris | *metadata:* This is the iris dataset, perhaps the best known database to be found in the pattern recognition literature. Fisher's paper is a classic in the field and is referenced frequently to this day. (See Duda & Hart, for example.) The data set contains 3 classes of 50 instances each, where each class refers to a type of iris plant. One class is linearly separable from the other 2; the latter are NOT linearly separable from each other. Predicted attribute- class of iris plant- setosa, versicolor, virginica.
*classes:* [setosa, versicolor, virginica]
*summary directive:* Tl;dr
*inference directive:* Based on the above information, predict if this flower will be classified as setosa, versicolor, virginica |
| tae | *metadata:* The data consist of evaluations of teaching performance over three regular semesters and two summer semesters of 151 teaching assistant (TA) assignments at the Statistics Department of the University of Wisconsin-Madison. The scores were divided into 3 roughly equal-sized categories ("low", "medium", and "high") to form the class variable.
*classes:* [low, medium, high]
*summary directive:* Tl;dr
*inference directive:* Predict whether this class will score low or medium or high: |
| glass | *metadata:* This is the glass dataset from USA Forensic Science Service; 6 types of glass; defined in terms of their oxide content (i.e. Na, Fe, K, etc). The study of classification of types of glass was motivated by criminological investigation. At the scene of the crime, the glass left can be used as evidence...if it is correctly identified!
*classes:* [building_windows_float_processed, building_windows_non_float_processed, vehicle_windows_float_processed, containers, tableware, headlamps]
*summary directive:* Tl;dr
*inference directive:* There are 6 possible type of glass: building_windows_float_processed, building_windows_non_float_processed, vehicle_windows_float_processed, containers, tableware, headlamps. Predict which one will this sample be: |
| breast-cancer | *metadata:* This is one of three domains provided by the Oncology Institute that has repeatedly appeared in the machine learning literature. This data set includes 201 instances of one class and 85 instances of another class. The instances are described by 9 attributes, some of which are linear and some are nominal. It contains information about women that had a recurrence or non-relapse of breast cancer after their first time.
*classes:* [recurrence, non-relapse]
*summary directive:* Based on the above examples, figure out under what conditions will a woman have recurrence or non-relapse of breast cancer?
*inference directive:* Predict whether this woman will have a recurrence or non-relapse: |
| visualizing-environmental | *metadata:* This is the visualizing-environmental dataset, one of the 22 data sets from the book Visualizing Data published by Hobart Press (books@hobart.com). This data describes indicators for a positive/negative environment based on ozone, radiation and temperature. *classes:* [positive, negative]
*summary directive:* Tl;dr
*inference directive:* There are clear signs of this environment being (positive or negative): |
| analcatdata-chlamydia | *metadata:* This chlamydia dataset is one of the data sets used in the book "Analyzing Categorical Data" by Jeffrey S. Simonoff, Springer-Verlag, New York, 2003. It contains results of individuals that tested for chlamydia.
*classes:* [positive, negative]
*summary directive:* Tl;dr
*inference directive:* Predict if this person will test positive or negative for chlamydia: |
| wine | *metadata:* This is the Wine recognition data. Updated Sept 21, 1998 by C.Blake. It contains results of a chemical analysis of wines grown in the same region in Italy but derived from three different cultivars. The analysis determined the quantities of 13 constituents found in each of the three types of wines.
*classes:* [1, 2, 3]
*summary directive:* Using these examples and based on the contents of constituents, summarize what distinguishes wines of type 1 or 2 or 3?
*inference directive:* Hence this wine will be classified as ->type |
| blood-transfusion-center | *metadata:* Data taken from the Blood Transfusion Service Center in Hsin-Chu City in Taiwan - this is a classification problem. The goal is to predict whether a given individual will consent or avoid donating blood.
*classes:* [consent, avoid]
*summary directive:* Tl;dr
*inference directive:* Therefore, this individual is likely to (avoid/consent): |
| somerville-happiness-survey | *metadata:* This is the Somerville Happiness Survey Data Set. It has ratings collected from a survey of Somerville residents. From the responses of a resident, the goal is to predict whether they feel happy or unhappy about the place.
*classes:* [unhappy, happy]
*summary directive:* Based on the Somerville happiness survey, how can we predict whether a resident is happy or unhappy with their place?
*inference directive:* So this resident is (happy or unhappy): |
| vehicle | *metadata:* This is the Statlog (Vehicle Silhouettes) Data Set. The purpose is to classify a given silhouette as one of four types of vehicle - bus, saab, opel or a van, using a set of features extracted from the silhouette. The vehicle may be viewed from one of many different angles.
*classes:* [bus, saab, opel, van]
*summary directive:* Using these examples, summarize how can we differentiate if a silhouette is that of a bus, saab, opel or a van.
*inference directive:* Out of saab, bus, van and opel, this vehicle is likely to be a |
| statlog-heart | *metadata:* This dataset is a heart disease database similar to a database already present in the repository (Heart Disease databases) but in a slightly different form. It has data on individuals having and not having heart disease.
*classes:* [present, absent]
*summary directive:* Differentiate people with heart disease present from ones absent.
*inference directive:* In this case, heart disease is likely to be (present/absent): |

| | |
|---|---|
| verterbra-column | *metadata:* This dataset contains values for six biomechanical features used to classify orthopaedic patients into 3 classes (normal, disk hernia or spondilolysthesis) or 2 classes (normal or abnormal). Biomedical data set built by Dr. Henrique da Mota during a medical residence period in the Group of Applied Research in Orthopaedics (GARO) of the Centre Médico-Chirurgical de Réadaptation des Massues, Lyon, France. The task is to classify patients as belonging to one out of two categories: Normal (100 patients) or Abnormal (210 patients).
*classes:* [abnormal, normal]
*summary directive:* Based on the above examples, summarize how will you distinguish patients that have normal vs. abnormal vertebral column.
*inference directive:* Therefore, this individual's vertebral column is likely to be (abnormal or normal): |
| ecoli | *metadata:* This data contains protein localization sites. Reference: "A Knowledge Base for Predicting Protein Localization Sites in Eukaryotic Cells", Kenta Nakai & Minoru Kanehisa, Genomics 14:897-911, 1992.
*classes:* [1, 2]
*summary directive:* Using these examples, how can we tell apart cells with protein localized in sites 1 and 2?
*inference directive:* Hence protein localization will be at site -> |
| haberman-survival | *metadata:* The dataset contains cases from a study that was conducted between 1958 and 1970 at the University of Chicago's Billings Hospital on the survival of patients who had undergone surgery for breast cancer.
*classes:* [survived, died]
*summary directive:* Based on these examples, figure out what commonalities are predictive of patients surviving more than 5 years and less.
*inference directive:* So, 5 years down the line, this person (survived/died): |
| diabetes | *metadata:* This dataset is originally from the National Institute of Diabetes and Digestive and Kidney Diseases. The objective is to predict based on diagnostic measurements whether a patient has high/low risk of developing diabetes.
*classes:* [low, high]
*summary directive:* Based on these examples, distinguish patients having low vs. high risk of diabetes.
*inference directive:* Based on the reasoning, this patient is likely to have a (low/high): |
| visualizing-hamster | *metadata:* This is the visualizing-hamster dataset contains 22 data sets from the book Visualizing Data published by Hobart Press (books@hobart.com). It contains examples of hamsters that are ill and healthy.
*classes:* [ill, healthy]
*summary directive:* Using these examples, identify predictive indicators of ill and healthy hamsters.
*inference directive:* Predict whether this hamster will be ill or healthy: |
| wholesale-customes | *metadata:* The data set refers to clients of a wholesale distributor. It includes the annual spending in monetary units (m.u.) on diverse product categories. This data gives information about spending patterns and region of operations of Retail and Horeca (Hotel/Restaurant/Café) customers of the wholesale distributor.
*classes:* [retail, horeca]
*summary directive:* Using these examples, summarize how can we differentiate Retail customers and Horeca customers.
*inference directive:* Therefore, which one of Retail or Horeca this customer is likely to be: |

Table 4: Continuous variable transformations applied to an example from the `wholesale-customers` dataset. The raw tabular record is as follows: spending on fresh products: 6353.0, spending on milk products: 8808.0, spending on grocery products: 7684.0, spending on frozen products: 2405.0, spending on detergents and paper products: 3516.0, spending on delicatessen products: 7844.0 and customer's region: Outside Lisbon and Porto.

| Method | Data Representation | Example as text |
|---|---|---|
| 4 bins + quantifiers {very low, low, high, very high} | - *spending on fresh products :* low
- *spending on milk products :* very high
- *spending on grocery products :* high
- *spending on frozen products :* high
- *spending on detergents and paper products :* high
- *spending on delicatessen products :* very high
- *customer's region :* Outside Lisbon and Porto | This customer spends low amounts on fresh products, very high amounts on milk products, high amounts on grocery products, frozen products, detergents and paper products, and very high amounts on delicatessen products. They are located outside of Lisbon and Porto. |
| 5 bins + quantifiers {very low, low, medium, high, very high} | - *spending on fresh products :* medium
- *spending on milk products :* very high
- *spending on grocery products :* high
- *spending on frozen products :* high
- *spending on detergents and paper products :* high
- *spending on delicatessen products :* very high
- *customer's region :* Outside Lisbon and Porto | This customer from outside Lisbon and Porto spends medium on fresh products, very high on milk products, high on grocery products, high on frozen products, high on detergents and paper products, and very high on delicatessen products. |
| 7 bins + quantifiers {extremely low, very low, low, medium, high, very high, extremely high} | - *spending on fresh products :* low
- *spending on milk products :* very high
- *spending on grocery products :* high
- *spending on frozen products :* high
- *spending on detergents and paper products :* very high
- *spending on delicatessen products :* extremely high
- *customer's region :* Outside Lisbon and Porto | This customer situated outside Lisbon and Porto spends low on fresh products, very high on milk products, high on grocery products, high on frozen products, very high on detergents and paper products, and extremely high on delicatessen products. |
| 9 bins + quantifiers {lowest, extremely low, very low, low, medium, high, very high, extremely high, highest} | - *spending on fresh products :* low
- *spending on milk products :* extremely high
- *spending on grocery products :* high
- *spending on frozen products :* high
- *spending on detergents and paper products :* very high
- *spending on delicatessen products :* highest
- *customer's region :* Outside Lisbon and Porto | This customer spends low amounts on fresh products, extremely high amounts on milk products, high amounts on grocery products, frozen products, detergents and paper products, and highest amounts on delicatessen products. They are located outside Lisbon and Porto. |
| 10 bins | - *spending on fresh products :* falls in the first out of ten bins of values.
- *spending on milk products :* falls in the second out of ten bins of values
- *spending on grocery products :* falls in the first out of ten bins of values
- *spending on frozen products :* falls in the first out of ten bins of values
- *spending on detergents and paper products :* falls in the first out of ten bins of values
- *spending on delicatessen products :* falls in the second out of ten bins of values
- *customer's region :* Outside Lisbon and Porto | This customer spends relatively little on fresh, grocery, frozen and detergents/paper products, and more on milk and delicatessen products. They are based outside Lisbon and Porto. |
| Percentile | - *spending on fresh products :* falls in the forty-first percentile
- *spending on milk products :* falls in the eighty-second percentile
- *spending on grocery products :* falls in the sixty-fifth percentile
- *spending on frozen products :* falls in the sixty-third percentile
- *spending on detergents and paper products :* falls in the seventy-second percentile
- *spending on delicatessen products :* falls in the ninety-eighth percentile
- *customer's region :* Outside Lisbon and Porto | This customer has an annual spending of 41st percentile on fresh products, 82nd percentile on milk products, 65th percentile on grocery products, 63rd percentile on frozen products, 72nd percentile on detergents and paper products, and 98th percentile on delicatessen products, and is located outside of Lisbon and Porto. |
| Standard deviation | - *spending on fresh products :* is within one std-dev below the mean value
- *spending on milk products :* is within one std-dev above the mean value
- *spending on grocery products :* is within one std-dev below the mean value
- *spending on frozen products :* is within one std-dev below the mean value
- *spending on detergents and paper products :* is within one std-dev above the mean value
- *spending on delicatessen products :* is two std-dev above the mean value
- *customer's region :* Outside Lisbon and Porto | The customer has annual spending on fresh products, milk products, grocery products, frozen products, detergents and paper products, and delicatessen products within one standard deviation of the mean, except for delicatessen products which is two standard deviations above the mean. The customer is located outside Lisbon and Porto. |
| Quartiles | - *spending on fresh products :* is between the first quartile and median values
- *spending on milk products :* is more than the third quartile value
- *spending on grocery products :* is between median and third quartile values
- *spending on frozen products :* is between median and third quartile values
- *spending on detergents and paper products :* is between median and third quartile values
- *spending on delicatessen products :* is more than the third quartile value
- *customer's region :* Outside Lisbon and Porto | This customer spends more than the third quartile value on milk, delicatessen and detergents and paper products. The customer's spending on fresh, grocery, and frozen products falls between the median and third quartile values, while the customer is located outside of Lisbon and Porto. |

Table 5: **Templatized descriptions:** Templates used to format examples for the ablation study between LLM-created data descriptions vs. template descriptions

| Dataset | Descriptive attribute values | Template |
|---|---|---|
| caesarian | *age:* [very young, young, middle-aged, old, very old]
*delivery_number:* [first, second, third, fourth, fifth]
*delivery_time:* [timely, premature, latecomer]
*blood_pressure:* [low, normal, high]
*heart_problem:* [has, doesn't have]
*delivery_mode:* [normal, caesarian] | This *{age}* woman is in her *{delivery_number}* delivery and it is *{delivery_time}*. She has a *{blood_pressure}* blood pressure and *{heart_problem}* heart problems. ### Based on these attributes, this woman is likely to deliver by *{delivery_mode}* |
| iris | *sepal_length*, *petal_length*: [very short, short, medium length, long, very long]
*sepal_width*, *petal_width*: [very narrow, narrow, medium width, wide, very wide]
*flower_type:* [setosa, versicolor, virginica] | This iris flower has *{sepal_length}* and {sepal_width} sepals. It also has *{petal_length}* and *{petal_width}* petals. ### Hence this flower is a *{flower_type}* |
| vertebral-column | *pelvic_incidence*, *pelvic_tilt*, *lumbar_lordosis_angle*, *sacral_slope*, *pelvic_radius*, *grade_of_spondylolisthesis*: [very low, low, medium, high, very high]
*result:* [normal, abnormal] | This patient has a *{pelvic_incidence}* pelvic incidence, *{pelvic_tilt}* pelvic tilt, and *{lumbar_lordosis_angle}* lumbar lordosis angle, *{sacral_slope}* sacral slope, *{pelvic_radius}* pelvic radius and *{grade_of_spondylolisthesis}* grade of spondylolisthesis. ### As a result, the patient's vertebral-column is likely to be *{result}* |
| statlog-heart | age: [very young, young, middle-aged, old, very old]
*sex:* [male, female]
*chest_pain_type:* [asymptomatic, nonanginal pain, atypical angina, typical angina]
*bp, cholesterol, st_depression, heart_rate, num_major_vessels:* [very low, low, medium, high, very high]
*fasting_blood_sugar:* [high, low]
*electrocardiographic_results:* [having left ventricular hypertrophy, normal, having ST-T wave abnormality]
*slope_st_segment:* [flat, upsloping, downsloping]
*exercise_induced_angina:* [has, do not have]
*defect_type:* normal, reversible, fixed
*presence_of_heart_disease:* [present, absent] | This individual is a/an *{age}* *{sex}* with *{chest_pain_type}* chest pain, *{bp}* resting blood pressure, and *{cholesterol}* serum cholesterol. Their fasting blood sugar *{fasting_blood_sugar}* >120 mg/dl, they are *{electrocardiographic_results}* and a *{heart_rate}* maximum heart rate. They *{exercise_induced_angina}* exercise-induced angina, and have a *{st_depression}* ST depression induced by exercise relative to rest. Their peak exercise ST segment has a *{slope_st_segment}* slope, and they have a *{num_major_vessels}* number of major vessels. The defect type is *{defect_type}*. ### Hence heart disease is likely to be *{presence_of_heart_disease}*. |
| haberman-survival | *age_at_time_of_op:* [very young, young, middle-aged, old, very old]
*year_of_op:* [1964, 1962, 1965, 1959, 1958, 1960, 1966, 1961, 1967, 1963, 1969,1968]
*num_pos_axillary_nodes:* [very low, low, medium, high, very high]
*survival_status:* [survived, died] | This patient was *{age_at_time_of_op}* at the time of operation in *{year_of_op}*. They had a *{num_pos_axillary_nodes}* number of positive axillary nodes detected. ### Therefore 5 years down the line, the patient *{survival_status}* |

Table 6: Comparing test error rate of `Summary Boosting` backended by Curie and ChatGPT on all datasets (↓). Refer to caption of Table 1 for the notations used.

| Dataset | Data Type | Size | Curie | ChatGPT |
|---|---|---|---|---|
| caesarian [cae] (42901) | 1c4d | 80 | $0.300_{\pm 0.04}$ | $0.406_{\pm 0.03}$ |
| iris (61) | 4c0d | 150 | $0.193_{\pm 0.03}$ | $0.083_{\pm 0.01}$ |
| tae (48) | 1c4d | 151 | $0.454_{\pm 0.03}$ | $0.443_{\pm 0.04}$ |
| glass (41) | 9c0d | 214 | $0.370_{\pm 0.02}$ | $0.492_{\pm 0.02}$ |
| breast-cancer [bc] (13) | 7c5d | 277 | $0.288_{\pm 0.02}$ | $0.360_{\pm 0.01}$ |
| visualizing-environmental [ve] (678) | 3c0d | 111 | $0.268_{\pm 0.03}$ | $0.333_{\pm 0.04}$ |
| analcatdata-chlamydia [ac] (535) | 2c2d | 100 | $0.170_{\pm 0.01}$ | $0.300_{\pm 0.06}$ |
| wine (43571) | 13c0d | 178 | $0.320_{\pm 0.01}$ | $0.250_{\pm 0.01}$ |
| blood-transfusion-center [btc] (1464) | 4c0d | 748 | $0.240_{\pm 0.04}$ | $0.433_{\pm 0.01}$ |
| somerville-happiness-survey [shs] [Koczkodaj, 2018] | 0c7d | 143 | $0.350_{\pm 0.02}$ | $0.430_{\pm 0.02}$ |
| vehicle (54) | 18c0d | 846 | $0.410_{\pm 0.04}$ | $0.350_{\pm 0.16}$ |
| statlog-heart [stath] [Dua and Graff, 2017] | 6c7d | 270 | $0.430_{\pm 0.01}$ | $0.370_{\pm 0.17}$ |
| verterbra-column [vc] (1524) | 6c0d | 310 | $0.262_{\pm 0.03}$ | $0.669_{\pm 0.03}$ |
| ecoli (1011) | 7c0d | 336 | $0.270_{\pm 0.03}$ | $0.193_{\pm 0.03}$ |
| haberman-survival [hs] (43) | 3c0d | 306 | $0.250_{\pm 0.01}$ | $0.415_{\pm 0.03}$ |
| diabetes [dia] (37) | 8c0d | 768 | $0.344_{\pm 0.01}$ | $0.297_{\pm 0.04}$ |
| visualizing-hamster [hams] (708) | 5c0d | 73 | $0.207_{\pm 0.00}$ | $0.400_{\pm 0.08}$ |
| wholesale-customers [wc] (1511) | 6c1d | 440 | $0.330_{\pm 0.00}$ | $0.199_{\pm 0.04}$ |

Table 7: Test error rate of `Summary Boosting` applied to Claude-2 vs. Curie ($\downarrow$). The experiments with Claude-2 were run using the same prompts as Curie from table 3.

| Dataset | Data Type | Size | Claude-2 | Curie |
|---|---|---|---|---|
| caesarian [cae] (42901) | 1c4d | 80 | $0.312_{\pm 0.03}$ | $0.300_{\pm 0.04}$ |
| iris (61) | 4c0d | 150 | $0.200_{\pm 0.02}$ | $0.193_{\pm 0.03}$ |
| breast-cancer [bc] (13) | 7c5d | 277 | $0.385_{\pm 0.05}$ | $0.288_{\pm 0.02}$ |
| vertebra-column [vc] (1524) | 6c0d | 310 | $0.310_{\pm 0.03}$ | $0.262_{\pm 0.01}$ |

