# OpenReview forum: "Language Models are Weak Learners"
_NeurIPS.cc/2023/Conference — NeurIPS 2023 poster_

### Official Review · Reviewer_im25 · 2023-07-01

**Soundness:** 3 good
**Presentation:** 3 good
**Contribution:** 3 good
**Rating:** 7
**Confidence:** 4

**Summary:**

This paper explored an interesting problem that how to apply and extend LLMs over tabular supervised learning tasks. The paper first described each tabular sample as text, and then resorted to LLM to generate the summary for a set of selected representative samples as the template, which can be viewed as a weak classifier. Finally, by integrating these induced weak classifiers followed by booting learning paradigm, a stronger boosting classifier is constructed. Experiments were conducted over several tabular classification data sets to demonstrated the effectiveness of such learning procedure.

This procedure makes full use of zero-shot/few-shot learning ability of LLMs and extends such ability to few-shot learning setting over tabular data.

**Strengths:**

1.This paper extends the application scope of LLMs to traditional tabular data and sheds light on integrating LLMs to many real world machine learning systems.
2.The paper gave some effective guidelines to transform tabular samples to text description based on LLMs and metadata automatically with minimal manual engineer.
3.Extensive experiments are conducted to demonstrate some important impact factors on the classification performance.

**Weaknesses:**

1.The current method seems not easy to apply on high-dimensional tabular data (and for high-dimensional data, we might suffer from insufficient training data, also might due to the limited text sequence handled by LLMs) or when there might be many irrelevant features (this might introduce noise in text description).

2.There are generally three types of features in traditional tabular data, quantitative, ordinal and categorical. The paper talks more about numerical features. It is better to give a systematical discussion of how to generate text description over these three different types while considering the meta-data or give the guidelines to generate the prompt patterns over LLMs to get the text description.

**Questions:**

1.In Algorithm 1, the proposed Clustering sampling is based on GPT embedding to group the data. How does the clustering performance? Such as inter- intra clustering distance metrics? Or does such clustering method will group samples with different class labels together? Can we also make use of some encoder based LLMs (such as BERT, BigBird for long sequence) to perform such clustering?
2.How to make use of meta-data of the feature to generate the text description of categorical feature values? And how to generate the summarization prompt pattern based on meta-data? Dose this pattern depend on the property of learning target? For example, binary classification vs multi-class classification.
3.Can we make use of results of tree-based method (such as XGBoost) to guide the preprocess of continuous features? That is, making use of same bin method with XGBoost and remap each bin to a meaningful text phrase based on the feature’s meta-data? This is due to the fact that the performance of proposed method heavily depends on such discretization method.
One benefit of XGBoost is that it is able to perform feature selection when generate sub-trees. The irrelevant features of course have negative impact on clustering and summarization. This might be also affecting the conclusion that more examples do not means better performance in “How does the performance scale with more examples?”

**Limitations:**

1.The proposed method might difficult to be directly applied to high-dimension tabular data.
2.The tabular data might be derived from different domains, such as biological? May be we need domain-specific fine-tuned LLMs or pretrained LM to perform clustering or summary to get optimal performance. Therefore, some conclusions especially from ablation study might be LLM depended.

---

> ### Author Rebuttal · Authors · 2023-08-10
>
> We are thankful for your positive review of our work! We are happy that you quote our experiments as extensive and as demonstrating the important aspects of our method. Please find our responses to your comments as follows.
>
> > 1. The current method seems not easy to apply on high-dimensional tabular data (and for high-dimensional data, we might suffer from insufficient training data, also might due to the limited text sequence handled by LLMs).
>
> Thank you for the comments! In general, high-dimensional data usually causes problems for machine learning due to the curse of dimensionality. So, for very high-dimensional data, we could generally expect there to be low-dimensional structure in the data. In our case, one could perhaps first summarize the individual data description to get a shorter data description. As we have explained in Appendix A.1, our data-to-text conversion pipeline tries to obtain a textual description of 80 words or less, which maintains a uniform length among all the descriptions.
>
> > 2. There are generally three types of features: quantitative, ordinal and categorical in tabular data. It is better to discuss/give guidlines on how to generate text description over these three different types while considering the meta-data.
>
> Currently, we encode only the quantitative features separately and we rely on the LLM to convert ordinal / categorical data appropriately. This is because we believe that the LLM should be able to infer what these types of data represent from the context and meta data.
>
> However, we could potentially prompt the model to be aware of these ordinal/categorical features explicitly. We do not investigate this direction in the current work, but we will add this for future work.
>
> **Questions:**
> > 1. In Algorithm 1, the proposed Clustering sampling is based on GPT embedding to group the data. How does this group samples with different class labels? What are distance metrics used? Can we also use some encoder based LLMs (such as BERT, BigBird for long sequence) for clustering?
>
> Specifications of our sampling are explained in Appendix A.7. The clustering algorithm used is Agglomerative Hierarchical Clustering (AGNES) from sklearn, with cosine distance metric, average linkage and a distance threshold of 0.05.
>
> Algorithm 2 presents our stratified cluster sampling approach. This method creates separate clusters for different class labels and independently samples from them in a stratified fashion. Specifically, for each class, we obtain samples uniformly over its respective clusters. The stratification is performed at the class-level, ensuring that the proportion of class labels in the obtained samples remains the same as in the original training dataset.
>
> Yes, we can also use other language embedding techniques such as BERT for getting these embeddings.
>
> > 2. How to make use of meta-data of the feature to generate the text description of categorical feature values? And how to generate the summarization prompt pattern based on meta-data? Dose this pattern depend on the property of learning target? For example, binary classification vs multi-class classification.
>
> We are not sure if we understand the question properly, but the meta-data is generally for the whole dataset. It is integrated into the prompts for data conversion and summarization, providing the LLM with contextual understanding of the task (refer to figures 1 and 2). Further explanation can be found in Appendix A.1-2.
>
> It's important to note that the inclusion of meta-data is independent of the target labels. Prompts for all datasets are detailed in Table 3. The summarization process is solely guided by the summarization directive, whether it's "Tl;dr" (Too long; didn't read) or "Summarize in detail." When making inference, the prompt does mention the target labels.
>
> > 3. Can we use XGBoost to guide the preprocessing of continuous features? That is, apply the same bin method with XGBoost and remap each bin to a meaningful text phrase based on the feature’s meta-data? The irrelevant features have a negative impact on clustering and summarization. This might be also affecting the conclusion that more examples do not means better performance in "How does the performance scale with more examples?"
>
> We thank the reviewer for this suggestion. This should be a promising experiment to try. Since trees created inside XGBoost have different feature splits in each boosting round, it might be difficult to formulate how they can be effectively combined to create a general discretization method for every feature. We leave this for future work.
>
> On the flip side, as we explore ways to improve the processing of numerical values, this problem itself might go away as newer LLMs may become better at quantitative reasoning.
>
> We further seek one clarification from the reviewer, to understand what they meant by: "This might be also affecting the conclusion that more examples do not means better performance in "How does the performance scale with more examples?""
>
> **Limitations:**
>
> > 1.The proposed method might difficult to be directly applied to high-dimension tabular data.
>
> We refer to our explanations addressed in the previous comments.
>
> > 2. The tabular data might be derived from different domains, such as biological? Can we use domain-specific fine-tuned LLMs or pretrained LM to perform clustering or summary ?
>
> The reviewer’s observation may be correct since tabular datasets can represent different domains. Indeed, we expect domain specific LLMs to perform better however we don't think is this a limitation per se. For example, in scenarios such as medical diagnosis, it can be more prudent to use a LLM pretrained on medical data which will achieve better results. For example a medical-specific LLM might perform better on these datasets - verterbra-column, breast-cancer, caesarian, blood-transfusion-center, and haberman-survival. This observation should be true of the LLM embeddings used in clustering as well.

---

> > ### Comment · Reviewer_im25 · 2023-08-17
> >
> > Thanks for detailed explanation and clarification on my questions. The response answers most of questions. And I keep my positive rating.

---

> > > ### Author Response · Authors · 2023-08-18
> > >
> > > Thank you for acknowledging our rebuttal and engaging in discussion!

---

### Official Review · Reviewer_gzhs · 2023-07-04

**Soundness:** 2 fair
**Presentation:** 3 good
**Contribution:** 2 fair
**Rating:** 6
**Confidence:** 4

**Summary:**

This paper explores the concept of weak learners, which are classifiers that achieve slightly better than random performance on any given data distribution. The paper demonstrates the effective utilization of large language models (LLMs) as weak learners. The study focuses on applying a large language model to tabular data using a boosting algorithm. By providing properly sampled text descriptions of tabular data samples according to the target distribution, LLMs can generate a summary or template for classification, serving as a weak learner for the task. The paper incorporates these models into a boosting approach, which, in certain cases, outperforms traditional tree-based boosting methods by leveraging the knowledge within the LLMs. The experimental results indicate that the proposed method outperforms both few-shot learning and complex fine-tuning procedures, particularly when dealing with a limited number of data points. These findings highlight the potential of prompt-based LLMs not only as few-shot learners themselves but also as components of larger machine learning pipelines. Overall, the paper showcases the effectiveness of prompt-based LLMs as weak learners in boosting algorithms for tabular data, offering insights into how they can improve classification performance, particularly in situations with scarce data availability.

**Strengths:**

Strengths:
---------------

1. The paper successfully brings together the concept of weak learners in boosting algorithms with the advancements in large language models (LLMs), creating a novel approach for utilizing LLMs as weak learners in tabular data classification.


2. The paper introduces a unique approach by converting tabular data into text form and using LLMs to generate summaries or prompts. This methodology allows the LLM-generated prompts to serve as effective templates for tabular data classification without the need for retraining or fine-tuning the LLM itself.


3. Through comprehensive evaluations, the paper demonstrates that the proposed approach outperforms alternative techniques such as zero-shot and few-shot learning. It also showcases the approach's superiority over traditional tree-based boosting and LLM-based fine-tuning methods, particularly in domains with limited examples. This performance advantage highlights the potential of LLMs as weak learners in boosting frameworks.


Overall the paper is well-written and paves a way to utilize LLMs in boosting.

**Weaknesses:**

Weaknesses:
------------------

1. The paper focuses specifically on tabular data classification, which may restrict the generalizability of the proposed approach to other types of data or domains. It would be valuable to explore the performance and applicability of LLM-based weak learners in a wider range of datasets and tasks.


2. Although LLMs have shown impressive performance in various natural language domains, they still have inherent limitations, such as sensitivity to input phrasing and potential biases in the training data. The paper would benefit from discussing and addressing these limitations to provide a more balanced perspective on the capabilities and potential drawbacks of LLM-based weak learners.
What was the rationale for choosing these datasets? Are there some data settings that are favorable to summary boosting and similarly the settings that are not good for the proposed method?

3. Minor comments/questions,

   a)  Please make fonts bold (or use color) in the tables for the best numbers in each row.

   b) Are the results reproducible? What is the cost of running the experiments?

**Questions:**

Please see the weaknesses above.

**Limitations:**

Yes.

---

> ### Author Rebuttal · Authors · 2023-08-10
>
> Thank you for your positive review of our manuscript! We appreciate your recognition of our work as novel and paving a way to utilize LLMs in boosting.
>
> **Weaknesses:**
>
> > 1. The paper focuses specifically on tabular data classification, which may restrict the generalizability of the proposed approach to other types of data or domains. It would be valuable to explore the performance and applicability of LLM-based weak learners in a wider range of datasets and tasks.
>
> Thank you for your suggestions! The primary goal of this paper is to test whether LLMs can be incorporated into larger ML systems like boosting. Since boosting is primarily used for tabular data, we decided to focus on tabular data. Indeed, in principle, the method can be applied to any other data in text form, such as GLUE, SQUAD and we plan to explore this in the future.
>
> > 2. The paper would benefit from discussing the limitations in LLMs such as sensitivity to input phrasing and biases in training data to provide a more balanced perspective on the capabilities and potential drawbacks of LLM-based weak learners.
>
> Indeed, LLM exhibit sensitivity to the presentation of prompts which elicit different behaviors. Although our core methodology involves using LLMs to generate prompts (both summary and inference), it still requires a modest amount of manual engineering. We have elaborately discussed failure modes in the creation of these LLM prompts in Appendix A.1-3. Moreover Table 3, lists all the prompts used in this paper, dataset-wise.
>
> The reviewer is correct in pointing out that any biases in the dataset will also reflect in the summary produced by the LLM and this is true of ML model in general.
>
> Yes our method inherits the problem of bias in LLM that comes from pretraining data. Specifically, it can affect the model’s ability to objectively summarize the examples or make predictions that are inconsistent with the biased pre-training data. We believe with better and debiased LLMs these issues can be alleviated. We will update the paper to emphasize more on this aspect.
>
> > 3. What was the rationale for choosing these datasets? Are there some data settings that are favorable to summary boosting and similarly the settings that are not good for the proposed method?
>
> These datasets were picked to be diverse in the number of features, proportions of different feature types (continuous, categorical) and the dataset size, so we believe that we have covered some of the most common settings. These were also the datasets commonly used in related papers, including TabPFN [1] and LIFT [2]. We haven’t tested on datasets with larger numbers of data points because the context size of current LLMs are limited.
>
> In Section 4.2 we have mentioned that our method generally works well for small tabular data without many continuous features. It especially works well when the task benefits from background knowledge where the LLM pretraining is helpful, such as *caesarian, somerville-happiness-survey, haberman-survival* and *TA evaluations*.
>
> When the dataset is large, this prior knowledge might become less relevant, so methods like finetuning become more competitive. Summary boosting is also less useful on datasets with many continuous variables, such as wine, iris, glass, vehicle, even though we encode these numerical values as descriptive attributes. Quantitative reasoning is inherently a problem with LLM but it is increasingly being solved in newer models like GPT-4 and Minerva.
>
> [1] N. Hollmann, S. Müller, K. Eggensperger, and F. Hutter. Tabpfn: A transformer that solves small tabular classification problems in a second. arXiv preprint arXiv:2207.01848, 2022.
>
> [2] T. Dinh, Y. Zeng, R. Zhang, Z. Lin, S. Rajput, M. Gira, J.-y. Sohn, D. Papailiopoulos, and K. Lee. Lift: Language-interfaced fine-tuning for non-language machine learning tasks. arXiv preprint arXiv:2206.06565, 2022.
>
> **Minor comments/questions:**
>
> > a) Please make fonts bold (or use color) in the tables for the best numbers in each row.
>
> We have posted a PDF of the table with the best-performing results highlighted in bold. We refer to our overall response note to all reviewers.
>
> > b) Are the results reproducible? What is the cost of running the experiments?
>
> Yes our results are reproducible and we have included code and instructions to replicate our experiments. However, there has been evidence that the ability of the models exposed through OpenAPI APIs are changing which is outside of our control.
>
> The cost of running the experiments is discussed in Appendix A.11.

---

> > ### Comment · Reviewer_gzhs · 2023-08-12
> > **Response to author rebuttal**
> >
> > I appreciate the authors' response and would like to share some lingering concerns regarding the utilization of OpenAI APIs for scientific research, primarily centered around issues of reproducibility, ethics, and cost implications. It might be worth considering an alternative title such as "Examining OpenAI GPT-3 as a Weak Learner" unless the scope of the study encompasses a broader range of language models, especially those that are openly accessible.
> >
> > While I find the concepts presented in the paper intriguing, I do have reservations about the evaluation methodology due to its reliance on a commercial API, which could potentially evolve over time. My intention is not to undermine the value of the paper's ideas, but rather to emphasize the importance of a robust evaluation process that stands up to scrutiny and remains valid regardless of potential changes in API availability.
> >
> > From a broader perspective, I am concerned about the overreliance on OpenAI APIs within machine learning research. Paying money to OpenAI to get better numerical results, and associating acceptance solely with better numerical results, can inadvertently limit the overall progress of the field.
> >
> > Personally, I would find the paper's concepts more compelling if the evaluation encompassed open language models that provide full transparency regarding their details and weights. My focus lies more on the soundness and reproducibility of the evaluation rather than pursuing impressive numerical results.

---

> > > ### Author Response · Authors · 2023-08-15
> > >
> > > Thank you for the response and engaging in discussion! We agree that the changing API is an important issue that needs to be addressed. **We will add more detailed discussion on this into our paper to emphasize the implication and limitation of using a commercial API**. On the other hand, as far as we are aware, the changes only affected ChatGpt instead of the model through API access so our results should not be affected. You should be able to reproduce the result with the code we provided.
> > >
> > > The reason why we chose OpenAI’s API access is not for numerical results. As you can see, the work is exploratory in nature and we do not actually always outperform existing methods. The actual reason is that using API lets us test the capabilities of the models at a reasonable efficiency even in an academic setting, as doing the boosting requires many sequential calls to the language model and we need to do it on many datasets in parallel. We fully agree that this is a less-than-ideal solution, but it does enable us to do research that would otherwise be inaccessible to us. At the current pace of innovation in LLMs, we are cautiously optimistic that there will be software solutions (e.g., for efficient inference) that will change the circumstances.
> > >
> > > Apologies that it took us a while to respond because we were investigating the possibility of using an open-sourced model. Unfortunately, as it stands, we simply do not have the infrastructure to host a LLM with size of GPT3 at a high-enough throughput to run all the experiments in a reasonable amount of time. We will try to reproduce subsets with open-source models such as BLOOM or GPT-J for the final version of the paper. In the meantime, is there anything else that we could discuss or add to the paper that would partially alleviate your concerns?

---

> > > > ### Author Response · Authors · 2023-08-21
> > > >
> > > > To address the concerns that our results are only based on GPT-3, we have investigated another LLM - Claude-2 and ran the Summary Boosting method for 4 datasets. The results are as follows.
> > > >
> > > > | **Dataset**        | **Claude-2 test error rate** | **Curie test error rate** |
> > > > |--------------------|------------------------------|----------------------|
> > > > |   iris             |   0.200                      |   0.193              |
> > > > |   caesarian        |   0.312                      |   0.300              |
> > > > |   vertebra-column  |   0.310                      |   0.262              |
> > > > |   breast-cancer    |   0.385                      |   0.288              |
> > > >
> > > > As you can see, the performance of Claude-2 is slightly worse than that of GPT-3 (but better than ChatGPT). Note that due to time constraints we have not tuned any prompts for Claude-2 so the actual performance is likely better. This experiment demonstrates that our method still works outside GPT-3, and is able to produce weak learners for different base LLMs. Though the scope of the experiments is limited, it shows preliminary evidence that the method is robust across models. Summarizing documents and text-based reasoning are pretty standard use cases of LLM, so in principle we expect our method to work with other prompt-based LLMs.
> > > > On the other hand, Claude-2 released by Anthropic is still accessed through a commercial API. We understand that you would like to see open-sourced models but as we mentioned above we do not have the resources to run experiments with these models. We will add the following discussion about the limitations of using commercial API in our revised paper:
> > > >
> > > > - **Limited Customization:** Commercial APIs may not allow you to fine-tune the model according to your specific needs. You might not have control over the architecture, hyperparameters, or training data used for the model, which can be crucial for certain research tasks. The backend models offered by these vendors are also continuously updated either through new training methods or more data. For ML research in general, it will be a good practice to mention the version of the model for replicating the experiments.
> > > >
> > > > - **Lack of Transparency:** The inner workings and architecture details of commercial APIs are often proprietary and not fully disclosed. This lack of transparency can make it challenging to understand model behavior, biases, and limitations. Specifically it deprives us of the opportunity to study the training process itself, which can be valuable for understanding how these models learn and for improving our own research methodologies. Commercial APIs may inherit biases present in the training data, potentially leading to biased or unfair responses.
> > > >
> > > > - **Data Privacy and Security:** Using a commercial API involves sending your data to an external server, which might raise concerns about data privacy and security, especially if you're dealing with sensitive or confidential information. There might also be regulatory and legal considerations when passing sensitive data to these API.

---

> > > > > ### Comment · Reviewer_gzhs · 2023-08-21
> > > > >
> > > > > Thank you for doing experiments with Claude-2 and sharing the results. It is understandable to not have the resources to run LLMs locally. I appreciate the discussion on limitations and it would be good to have it in the main paper.  I know it's quite late, but would it be possible to run these experiments on some of the quantized versions of LLMs available on Hugging Face? I hope the authors will include these in the main paper and make improvements in the presentation as suggested by other reviewers. I am happy to increase my score.

---

### Official Review · Reviewer_NJzp · 2023-07-05

**Soundness:** 3 good
**Presentation:** 3 good
**Contribution:** 2 fair
**Rating:** 4
**Confidence:** 4

**Summary:**

The paper investigates the use of large language models (LLMs) as weak learners in a boosting algorithm applied to tabular data. By providing text descriptions of tabular data samples, LLMs can generate a summary that acts as a template for classification, effectively serving as a weak learner. The authors incorporate these LLM-generated weak learners into a boosting approach and demonstrate their performance superiority over few-shot learning and fine-tuning procedures, especially for tasks with limited data points.The results showcase the potential of prompt-based LLMs not only as standalone learners but also as components of larger machine learning pipelines.

**Strengths:**

1. The experiments are conducted on a large number of datasets and illustrate the effectiveness of the method proposed on tabular data.
2. Clear writing and visualization.
3. The examples in the appendix are enjoyable to read. The prompt templates in the appendix are worth learning.

**Weaknesses:**

1. The integration of multiple weak learners using ensemble learning methods, each requiring the invocation of LLM, may result in significant resource costs. In Appendix A.11, it is calculated that running a dataset of 175 instances would incur a cost of $25, which seems relatively high.
2. The presentation of experimental results could be improved by highlighting the relevant information in the tables. Adding bold formatting to the results would make them more prominent and eye-catching.
3. In Appendix A.12, it is mentioned that ChatGPT performs poorly on some datasets due to the utilization of RLHF. However, no experimental evidence or persuasive argument is provided to support this claim. It is also possible that the issue stems from suboptimal prompts. Also, the Figure 3 shows the results of using GPT-3, where GPT-3 does not consistenty improve upon the smaller model. GPT-3 and ChatGPT have the same pretrained data, which means the performance degradation may be influenced by the pretraining data of the model.

**Questions:**

1. The proposed method to split continuous values in tabular data into discrete attributes and generate corresponding summaries. I wonder whether using this approach to fine-tune the LLM could yield promising results. This way, it can effectively align the tabular data format with the language model while enabling the model to learn latent knowledge embedded in the data.
2. How can the determination of the number of discrete categories (e.g., low, medium, high) for continuous numerical values in a table be made? From what I understand, your approach involves experimenting to determine the optimal number of categories.

**Limitations:**

Limitations are discussed in Section 6 of this paper.

---

> ### Author Rebuttal · Authors · 2023-08-10
>
> We are glad that you found our paper enjoyable to read and our writing lucid to follow. Thank you for your positive feedback of our work! Please find our responses to your review as follows.
>
> **Weaknesses**
>
> > 1. The integration of multiple weak learners using ensemble learning methods, each requiring the invocation of LLM, may result in significant resource costs. In Appendix A.11, it is calculated that running a dataset of 175 instances would incur a cost of $25, which seems relatively high.
>
> Thank you for the suggestion! It is true that querying OpenAI for creating the prompts seem to be expensive, however in principle our approach does not need OpenAI API and can utilize open-source models hosted locally which could drastically reduce the cost.
>
> We also show in Appendix A.10 that our method is infact more compute-efficient compared to finetuning. For a dataset of 175 points, finetuning makes 7000 calls to the LLM, while prompting needs only 1250 passes through the LLM.
>
> > 2. The presentation of experimental results could be improved by highlighting the relevant information in the tables. Adding bold formatting to the results would make them more prominent and eye-catching.
>
> We have posted a PDF of the table with the best-performing results highlighted in bold. We refer to our overall response note to all reviewers.
>
> > 3. In Appendix A.12, it is mentioned that ChatGPT performs poorly on some datasets due to the utilization of RLHF. However, no experimental evidence or persuasive argument is provided to support this claim. Also, GPT-3 does not consistenty improve upon the smaller model and GPT-3 and ChatGPT have the same pretrained data, which means the performance degradation may be influenced by the pretraining data of the model.
>
> In Appendix A.12, we show that the performance of ChatGPT is worse than GPT-3 Curie on many datasets, especially medical-related such as patients, diseases, etc.
>
> We speculate that RLHF is responsible for this behavior since it is one of the main differences between ChatGPT and GPT-3 Curie. Since we do not have information about the pretraining data changes, we believe it is reasonable to infer that RLHF is a major factor. We note that we did not alter any prompts for ChatGPT and our method mainly relies on the LLM generating summary for itself which it uses to perform reasoning.
>
> We have provided our explanations based on experiments in Table 6. We have observed that it was hard for models with RLHF to change their opinion on scenarios that sound unlikely but plausible. For example in the *somerville-happiness-survey* dataset, a resident rates many amenities as low but is overall *happy* due to other factors. ChatGPT was unable to reconcile this instance with the remaining examples. Further, we have observed that the models with RLHF often would abstain from answering (making predictions) on such examples, making the boosting algorithm less effective.
>
> Again, we wish to highlight that this is just speculation based on our empirical observations, since we don't have access to the actual models/ pretraining data of OpenAI. This observation is important because if one wishes to use this method it would be better to use the base pretrained model rather than one that has been fine-tuned with RLHF.
>
> **Questions:**
>
>
> > 1. The proposed method to split continuous values in tabular data into discrete attributes and generate corresponding summaries. I wonder whether using this approach to fine-tune the LLM could yield promising results. This way, it can effectively align the tabular data format with the language model while enabling the model to learn latent knowledge embedded in the data.
>
> Yes we have compared a similar approach called LIFT [1] in Table 2, which is comparable to XGboost on most datasets. It involves finetuning the LLM directly with plain English sentence form of the tabular records. Without needing to convert continuous values to discrete categories, this method is already competitive on datasets with many continuous features such as glass, wine, iris, vehicle, and wholesale-customers. This suggests that with finetuning, LLMs are able to handle continuous attributes better.
>
> [1] T. Dinh, Y. Zeng, R. Zhang, Z. Lin, S. Rajput, M. Gira, J.-y. Sohn, D. Papailiopoulos, and K. Lee. Lift: Language-interfaced fine-tuning for non-language machine learning tasks. arXiv preprint arXiv:2206.06565, 2022.
>
> > 2. How can the determination of the number of discrete categories (e.g., low, medium, high) for continuous numerical values in a table be made? From what I understand, your approach involves experimenting to determine the optimal number of categories.
>
> Yes in Appendix A.6 and Table 4 we provide a detailed description of various techniques used for encoding numerical values into descriptive attributes. Among these strategies, we found that binning with quantifiers works best, and further the the number of bins was determined through hyperparameter search. These results are summarized in Figure 4 (right top).

---

### Official Review · Reviewer_LBci · 2023-07-06

**Soundness:** 2 fair
**Presentation:** 2 fair
**Contribution:** 2 fair
**Rating:** 4
**Confidence:** 2

**Summary:**

This paper demonstrates that prompt-based LLMs can be used as weak learners, with applications on boosting algorithms for tabular data. By providing text descriptions of tabular data samples, the authors show that LLMs can produce a summary of the samples and use it as a template for classification that can be leveraged as weak learners for the task at hand. The proposed approach outperforms zero- and few-shot learning and, occasionally, even SOTA algorithms.

**Strengths:**

The paper introduces a novel approach to using LLMs as weak learners that can be leveraged on tabular tasks via boosting. The proposed approach appears to be novel, but in order to properly judge the paper's significance, the authors should address the comments below on the quality and clarity of presentation. Compared to zero and few- shot approaches, the proposed Summary & Summary Boosting is clearly superior; however, when compared with KNN, XGBoost, and (especially) TabPFN, the practical applicability seems quite limited and not well understood.

**Weaknesses:**

The paper can be significantly improved by providing a crisper, more intuitive description of the proposed approached (both Summary & Summary Boosting), and by offering an in-depth discussion of the practical impact of the empirical results in Section 4.

With respect to the presentation: Figure 2 appears to present the "Summary" algorithm, without Boosting. Even so, it is unclear (i) what is the output of the stratified cluster sampling (I expect it to be a subset of the input to this step), (ii) what does "Select BEST summary" mean - best based on what? only one summary is selected? probably not, because one would expect it to be not so much "the best" but rather "the MOST USEFUL" to classify a new, unlabeled example. Furthermore, lines 137-138 seem to imply that just one summary is chosen (per class?). Adding all these key details to Figure 2 is critical for removing the current ambiguities.

The paper should add a new figure, similar to Fig 2, that fully illustrates in detail the Summary Boost algorithm.

Last but not least, the authors should also clarify & discuss the results in Tables 1 & 2, for both of which they should BOLD the best result (to improve readability). While it is obvious that both Summary & Summary Boosting outperform zero- and few- shot learners (Table 1), the picture is a lot more confusing when comparing Summary Boosting with the four strong "baselines" in Table 2.  First of all, the novel approach obtains the best results on only 3 or 4 of the 16 datasets. Second, often times one of the baselines outperforms the novel approach by almost an order of magnitude (eg, wc & wine). Last but not least, the two conclusions of the paper (that SB does best on datasets with few examples and worst on those with continuous attributes) are not necessarily correct: even though the best performances of SB are on datasets with 1/3/3/5 continuous attributes and 73-306 examples, it is unsafe to generalize from here. For example, iris has 4 continuous attributes and 150 examples, but SB's performance on it is abysmal (0.193 vs 0.027), even though this dataset's properties are similar in nature hams (5 & 73) or tae (1 & 151), where it does comparable/better than the competition. The authors should do an in-depth study about the suitability of the approach to various types of domains; if they need extra space, they could shrink/send-t0\o-appendices Section 5.

OTHER COMMENTS:
- line 87:    you should either explain what "the special token" does, or omit to mention it


**Questions:**

see Weakness above

**Limitations:**

The main concern of this reviewer is the practical usage of Summary & Summary Boost. Without clarifying the issues and questions raised under "Weaknesses," the proposed approach has very limited applicability under still-unclear circustances.

---

> ### Author Rebuttal · Authors · 2023-08-10
>
> Thank you for your valuable suggestions for our manuscript.
>
> > 1. The paper can be significantly improved by providing a more intuitive description of the proposed approachs and by discussing the practical impact of the results in Section 4.
>
> Thank you for your comments! We have brought out the intuiton behind our proposed method in multiple places leading upto Section 3.1. Lines 32-37 introduces the idea, lines 77-80 and 95-100 explain it further.
>
> A practical applicability of our work will be the fact it improves the ability of LLMs to reason on tabular data over few-shot and zero-shot. We have shown that “summary” as an intermediate step makes reasoning easier and infact can be used inside boosting models for small tabular data upto 300 data points without many continuous features.
>
> > 2. With respect to the presentation: Figure 2 appears to present the "Summary" algorithm, without Boosting. Even so, it is unclear (i) what is the output of the stratified cluster sampling
>
> Stratified cluster sampling yields a representative subset of examples that will be injected into the prompt for summarization. Since the LLM’s context length is limited, only a few examples selected by this sampling process can be fed as input. This has been mentioned in lines 142-154 and depicted in Figure 2.
>
> > 3. What does "Select BEST summary" mean - best based on what? only one summary is selected? probably not, because one would expect it to be not so much "the best" but rather "the MOST USEFUL" to classify a new, unlabeled example.
>
> We use validation set to select the best summary. The best summary is the one that achieves the least validation error rate.  We are not sure what “most useful” could mean in this case without a more concrete definition.
>
> > 4. Furthermore, lines 137-138 seem to imply that just one summary is chosen (per class?).
>
> We would like to clarify that there is only one summary for each weak learner.
>
> For the  "Summary" method compared in Section 4 - Experiments, we generate a fixed number (25) of summaries and pick the one with the smallest validation error rate. This is mentioned in lines 137-139.
>
> However to obtain a weak learner that is used inside "Summary Boosting", the procedure is slightly different. We sample summaries until we find the first summary that does better than random guessing on the training distribution in that round. This summary will be our weak learner. We refer to the lines 165-167.
>
> Further details have been explained in Appendix A.2.
>
> > 5. Adding all these details to Figure 2 is critical for removing the current ambiguities.
>
> We thank the reviewer for the suggestion. Figure with the boosting algorithm included will take up a lot of space. So, we found it better to describe the missing details in text and through Algorithm 2. In the revision we can refer to the text in the caption if that will be more helpful.
>
>
> > 6. The authors should also clarify & discuss the results in Tables 1 & 2, for both of which they should BOLD the best result (to improve readability). While both Summary & Summary Boosting outperform zero- and few- shot learners (Table 1), it is not true of the "baselines" in Table 2.
>
> We have posted a PDF of the table with the best-performing results highlighted in bold. We refer to our overall response note to all reviewers.
>
> > 7. The two conclusions of the paper (that SB does best on datasets with few examples and worst on those with continuous attributes) are not necessarily correct: iris has 4 continuous attributes and 150 examples, but SB's performance on it is abysmal (0.193 vs 0.027), even though this dataset's properties are similar in nature hams (5 & 73) or tae (1 & 151), where it does comparable/better than the competition.
>
> Our observations are based on the majority of datasets and not any single one. It is true that the LLM does not perform well when the dataset has more continuous features as supported by its error rate on *iris, glass, wine*, and *wholesale-customer* datasets. On the other hand, our approach works best when the dataset is small, as evident in *caesarian, TA evaluations, somerville-happiness-survey, haberman-survival*, and *visualizing-hamster* datasets. This is reasonable because LLM leverages prior knowledge from pretraining, which becomes less relevant and less competitive as the dataset size increases.
>
> However, it's worth noting that the *visualizing-hamster* dataset is very small (< 80 points), and in this case, it appears that the XGboost overfitting makes our method seem like the best performer. It is our hypothesis that prompting is likely to not overfit on any dataset since it remains a few-shot learner.
>
> We wish to highlight here that our method does not always show improvements over the state of the art due to some current limitations in LLM with respect to context size that prevent it from ingesting a large number of examples. This may be addressed in newer LLMs - for instance, GPT-4 has 32k context length. Additionally, even converting numerical values to discrete categories does not help address continuous features, which is a known problem in LLM that they are bad at quantitative reasoning. These issues may be resolved as LLMs get better. Also there may be ways to combine trees and LLM in a single boosting algorithm.
>
> > 8. line 87: you should either explain what "the special token" does, or omit to mention it
>
> By "special token" we mean the [MASK] token in prompt which is commonly used for Masked Language Modeling in BERT, i.e. predicting a masked token in a sequence. We will include this change in the revision.
>
> **Limitations:**
>
> > 1. The main concern of this reviewer is the practical usage of Summary & Summary Boost. Without clarifying the issues and questions raised under "Weaknesses," the proposed approach has very limited applicability under still-unclear circustances.
>
> We refer to our explanations addressed in the previous comments. We will emphasize these aspects more in the revised paper.

---

> > ### Comment · Reviewer_LBci · 2023-08-17
> > **After authors' rebuttal**
> >
> > Thank you very much for the detailed answer to my review. Your comments helped clear up most of my "tactical comments," but less so when it comes to the "strategic" ones. IMO, the empirical evidence is still too mixed to get this paper out of the "borderline" zone; the situation would be different if the authors had a real-world application domain with clear cut results. Furthermore, the comments of fellow reviewers NJzp and gzhs have also convinced me that my original rating is correct.

---

### Official Review · Reviewer_inEG · 2023-07-28

**Soundness:** 3 good
**Presentation:** 4 excellent
**Contribution:** 2 fair
**Rating:** 6
**Confidence:** 4

**Summary:**

This paper proposes a novel way to use large language models (LLMs) as weak learners in boosting frameworks for tabular data. The core idea is called LLM Summary Boosting, a novel method that prompts large language models (LLMs) to create weak learners for usages within a boosting framework to make predictions on tabular data. The authors conducted thorough experiments on a variety of benchmark datasets. Compared to conventional tree-based boosting and LLM-based finetuning, this approach demonstrates superior performance in specific scenarios and remains effective with limited examples. To the best of my knowledge, the contribution of this paper is novel despite its incremental nature of contribution. I therefore recommend weak accept and am willing to hear what the authors think.

**Strengths:**

This paper is well-motivated and well-written. I also find the paper very easy to follow along with. Ideas of prompting LLMs to perform data manipulation/wrangling task aren’t new but this paper focuses on the specific usages of prompting for use with boosting scheme on tabular data. I think the contribution is clear here with the experiments.

**Weaknesses:**

While I find the arguments and experiments quite comprehensive, I am personally on the fence about the title, but I do not fully object to it. However, in the current form, the title seems to be a little bit overclaiming.

**Questions:**

N/A

**Limitations:**

The authors described the limitations of the proposed methods in section 6. The authors also provide a detailed failure modes documentation and I think it is very valuable.

---

> ### Author Rebuttal · Authors · 2023-08-09
>
> Thank you for your positive views of our paper! We are glad that you found it well-written and easy to follow along with. Please find our responses to your review as follows.
>
> **Weaknesses**
>
> > 1. While I find the arguments and experiments quite comprehensive, I am personally on the fence about the title, but I do not fully object to it. However, in the current form, the title seems to be a little bit overclaiming.
>
> Thank you for your suggestion! We have justified the title by showing that LLMs can be used for boosting, i.e. serve as weak learners.
>
> By definition, a weak learner is a classifier that achieves better than random guessing performance under the distribution of interest. We believe that our tabular data classifier created through prompts was able to demonstrate this property.

---

### Author Rebuttal · Authors · 2023-08-10

A common point shared by many reviewers was to make fonts bold in the tables 1 & 2 for the best numbers in each row. We have added bolding to highlight the best-peforming results which can be viewed in the attached file.

We note that our method doesn't always show improvements over the state of the art due to some current limitations in LLM with respect to context size that prevent it from ingesting a large number of examples (might be solved in GPT-4 or newer models). We show that our approach works best when the dataset is small and doesn't contain many continuous features, which is reasonable since LLM exploits prior from pretraining to its advantage. In the future, one may combine this strength with other models to get the best of both worlds.

---

### Decision · Program_Chairs · 2023-09-21

**Decision:**

Accept (poster)

**Comment:**

This paper asks the question: "Can large language models serve as weak learners?" Here, weak is used in the sense of boosting algorithms. Text descriptions of tabular data are fed into large language models (LLMs) and the resulting prompt responses are used within a boosting algorithm. This is an interesting way to combine prompts that can outperform typical few-shot learning.

The reviewers generally liked the creative combination of traditional machine learning and LLMs. This is a novel direction and, as the paper indicates, opens up interesting possibilities for incorporating LLMs into larger machine learning pipelines. The reviewers made several suggestions for improving the presentation of the paper, which the authors are strongly encouraged to include in the final version. One reviewer raised concerns regarding the cost of applying LLMs in this way, but as the authors say, this work points in a future direction that can also be explored with open-source models and the reviewer did not respond. While this approach might not be completely practical at this time, the approach seems promising and this direction should be shared with the community.